# When Model Merging Breaks Routing: Training-Free Calibration for MoE

**Canbin Huang** [1]  **Tianyuan Shi** [1]  **Xiaojun Quan** [1 2]  **Jingang Wang** [3]  **Jianfei Zhang** [3]  **Qifan Wang** [4]

## Abstract

Model merging has emerged as a cost-effective approach for consolidating the capabilities of multiple LLMs without retraining. However, existing merging techniques, largely based on linear parameter arithmetic or optimization, struggle when applied to Mixture-of-Experts (MoE) architectures. We identify a critical failure mode in MoE merging, termed *routing breakdown*, in which the merged router fails to dispatch tokens to suitable experts. Routing breakdown stems from the sensitivity of the non-linear softmax and discrete Top-$k$ routing mechanisms to parameter perturbations from merging, a sensitivity further amplified by load-balancing constraints imposed during MoE pretraining. Because fine-tuned experts exhibit distinct specializations, even modest misrouting can cause severe performance degradation. To address this issue, we propose Hessian-Aware Router Calibration (HARC), a training-free framework that leverages second-order curvature information to realign the merged router. This approach admits a closed-form solution that can be efficiently solved using a matrix-free conjugate gradient method. Experiments on mathematical reasoning and code generation tasks show that HARC effectively mitigates routing breakdown across diverse MoE merging baselines and leads to substantial performance improvements. Our code is available at https://github.com/huangcb01/HARC.

## 1. Introduction

Model merging has emerged as an effective approach for consolidating the capabilities of multiple task-specific large language models (LLMs) into a single unified model. This

[1]School of Computer Science and Engineering, Sun Yat-sen University, China [2]Shenzhen Loop Area Institute, China [3]Meituan, Inc., China [4]Meta AI, USA. Correspondence to: Xiaojun Quan <xiaojunquan@slai.edu.cn>.

*Proceedings of the 43rd International Conference on Machine Learning*, Seoul, South Korea. PMLR 306, 2026. Copyright 2026 by the author(s).

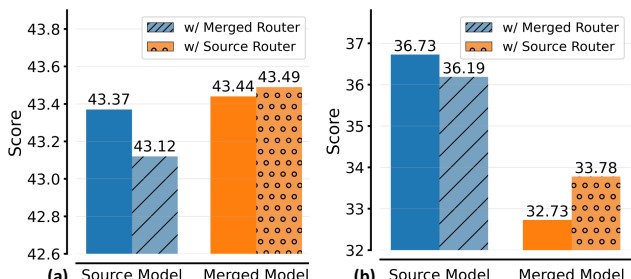

*Figure 1.* Illustration of routing breakdown using existing merging methods for MoE models. Cross-evaluation results on **(a)** mathematical and **(b)** code tasks demonstrate the impact of router swapping. Solid bars denote models operating with their default routers. Replacing the original router with the merged router (blue hatched bars) degrades the source model's performance, whereas restoring the source router to the merged model (orange dotted bars) significantly recovers its capability.

avoids the prohibitive costs associated with joint multi-task training and the inference overhead of ensemble methods. Techniques such as Task Arithmetic (Ilharco et al., 2023) and Ties-Merging (Yadav et al., 2023) have demonstrated that parameter-space arithmetic can successfully combine specialized knowledge. However, most existing research primarily focuses on dense architectures and the alignment of linear layers (Jin et al., 2025; Cheng et al., 2025).

As Mixture-of-Experts (MoE) architectures (Shazeer et al., 2017) gain wider adoption by enabling scalable parameter counts while preserving inference efficiency, the problem of merging sparse model components remains largely unexplored. Unlike dense models, MoE architectures rely on a routing module that dynamically dispatches tokens to a subset of experts. We argue that traditional linear model merging methods are ill-suited for this setting due to the inherent sensitivity of the routing mechanism. In particular, the router typically combines a softmax activation with a discrete Top-$k$ selection process, under which small perturbations in the router's weights can be exponentially amplified. This effect is further exacerbated by load-balancing objectives used during pretraining, which can amplify instability and lead to degenerate routing behavior after merging. As a consequence, even minor merging-induced errors may result in incorrect expert assignments, leading to substantial divergence in output representations even when the expert parameters themselves are preserved. This motivates the following research question: *Does model merging destabilize*

*routing in MoE architectures and degrade performance?*

To investigate this, we conducted a preliminary study by merging two OLMoE (Muennighoff et al., 2025) models fine-tuned for mathematics and code generation, utilizing the state-of-the-art WUDI (Cheng et al., 2025) method. By tracking token-level expert assignments across all layers, we uncovered a severe *expert mismatch* phenomenon: 50.11% of expert selections deviate from the source models. This instability compounds rapidly through the network depth; across 16 layers, 99.98% of tokens experience routing divergence, with an average of 8.02 mismatched decisions per token. To isolate routing mismatch as the primary driver of performance degradation, we performed a cross-evaluation by decoupling the routing module from the expert parameters. As illustrated in Figure 1, equipping the merged model with the oracle source router yields substantial performance recovery, whereas applying the merged router to the original source model consistently harms results. These findings underscore that preserving the experts' capabilities is futile without ensuring consistent routing behavior, highlighting the urgent need for a specialized merging scheme for routers.

To address this challenge, we propose **Hessian-Aware Router Calibration (HARC)**, a training-free strategy designed to match the router's output distributions before and after merging. This approach leverages a second-order Hessian approximation of the routing objective, enabling the derivation of a closed-form solution. Notably, HARC preserves the router's output distributions without requiring gradient backpropagation, significantly reducing computational overhead. By directly optimizing routing behavior, it ensures that the merged model maintains consistent expert selection logic across different tasks. For the remaining layers, we employ existing dense merging techniques, which have been shown to be effective for linear layers. These methods ensure that specialized knowledge in dense layers is properly transferred, while the router's complexity is handled by the proposed training-free strategy.

Empirical evaluations on the merging of math and code MoE models demonstrate that HARC effectively addresses the routing breakdown challenge, consistently improving over traditional model merging methods. Furthermore, our analysis highlights three key findings. *First*, we observe that routing errors accumulate more markedly in deeper layers of the MoE model, particularly as experts become more specialized, which can lead to catastrophic semantic drift. HARC mitigates this by incorporating Hessian curvature information, suppressing this trend and maintaining alignment with the original routing behavior, especially in these critical deeper layers. *Second*, our investigation into data composition reveals that using clean prompts for calibration outperforms using noisy or off-policy full text, suggesting that correctly routing tokens from the outset is crucial for

maintaining a robust and consistent routing trajectory. *Finally*, we demonstrate that HARC exhibits remarkable data efficiency, enabling the model to converge to near-optimal performance with around 40% of the calibration samples.

## 2. Related Work

**Parameter-Space Model Merging.** Most traditional model merging methods operate in parameter space via simple arithmetic operations. Model Soups (Wortsman et al., 2022) shows that averaging the weights of models fine-tuned under different hyperparameters can improve accuracy and robustness. Task Arithmetic (Ilharco et al., 2023) formalizes this idea by representing task adaptations as *task vectors* and composing them through vector arithmetic. A common issue is that naive averaging can degrade performance due to parameter interference and sign disagreement. To mitigate interference, TIES-Merging (Yadav et al., 2023) eliminates redundant parameters and resolves sign conflicts through a trim-elect-merge procedure, while DARE (Yu et al., 2024) sparsifies delta parameters via random dropping and rescaling. FuseChat (Wan et al., 2024) proposes the SCE (Select, Calculate, Erase) method, which derives fine-grained merging coefficients from the magnitude of parameter updates and eliminates conflicting update directions. DELLA-Merging (Deep et al., 2024) refines pruning through magnitude-based sampling, and TALL-masks (Wang et al., 2024) constructs binary masks to stitch task-relevant parameter supports while filtering task-irrelevant noise. AWD (Xiong et al., 2024) decomposes task vectors into shared and task-specific components to mitigate interference via disentanglement. Beyond heuristic filtering, DOGE (Wei et al., 2025) formulates merging as constrained optimization in a shared subspace solved by projective gradient descent, while WUDI-Merging (Cheng et al., 2025) leverages approximate subspace structure of task vectors to optimize an objective that reduces interference.

**Data-Driven Model Merging.** While parameter-space methods are efficient, they can miss functional interactions that emerge during inference. Data-driven approaches incorporate input-side statistics or behavioral signals to guide merging. Fisher Merging (Matena & Raffel, 2022) and Reg-Mean (Jin et al., 2025) reweight parameters using Fisher information or activation statistics. AdaMerging (Yang et al., 2024) learns task-specific coefficients via entropy minimization on unlabeled data. Activation-Informed Merging (Nobari et al., 2025) uses activation-space mutual information as a plug-and-play criterion for layer-wise coefficients. Sens-Merging (Liu et al., 2025) estimates sensitivity using validation data to balance parameter updates. NeuronMerge (Gu et al., 2025) clusters neurons by activation patterns before merging to better preserve diverse capabilities. Leveraging Submodule Linearity (Dai et al., 2025) further identifies

locally linear regions within specific submodules (e.g., attention or MLP) and performs merging within these regions.

**Limitations of MoE Merging.** Despite recent progress, existing techniques largely rely on the Linear Mode Connectivity (LMC) hypothesis (Frankle et al., 2020), which assumes that linear interpolation in parameter space yields smooth functional transitions. This assumption becomes brittle for Mixture-of-Experts (MoE) models, where inference critically depends on *routing*: tokens are dispatched via nonlinear softmax scores followed by discrete Top-$k$ selection, inducing inherently non-smooth behavior. As a result, even modest perturbations to router parameters during merging can trigger a failure mode we term *routing breakdown*, in which the merged router fails to consistently dispatch tokens to appropriate experts. This vulnerability is further exacerbated by load-balancing regularization used during MoE pretraining, which makes expert assignments highly sensitive to gating shifts. Since fine-tuned experts often encode specialized competencies, such misrouting can cause disproportionate and severe performance degradation. Collectively, these factors challenge the applicability of standard merging methods to MoE merging.

## 3. Preliminaries

In this section, we formalize the Mixture-of-Experts (MoE) routing mechanism and analyze the theoretical limitations of applying existing linear merging techniques to the inherently non-linear routing module.

### 3.1. MoE Routing Formulation

Consider a standard MoE layer consisting of $K$ experts $\{E_1, \ldots, E_K\}$ and a routing network. Let $\mathbf{x} \in \mathbb{R}^d$ represent the input representation, where $d$ is the hidden dimension. The routing module computes a probability distribution $\mathbf{r} \in \mathbb{R}^K$ over the experts using a learnable weight matrix $\mathbf{W} \in \mathbb{R}^{K \times d}$. The routing logits $\mathbf{z} = \mathbf{W}\mathbf{x}$ are passed through the softmax function to obtain the routing probabilities:

$$r^{(k)} = [\text{softmax}(\mathbf{z})]_k = \frac{e^{z^{(k)}}}{\sum_{j=1}^{K} e^{z^{(j)}}}. \quad (1)$$

In practice, MoE models activate only a subset of experts with the highest routing probabilities (e.g., Top-$k$ gating). The output of the MoE layer is thus computed as:

$$\mathbf{y} = \sum_{k \in \mathcal{T}} r^{(k)} E_k(\mathbf{x}), \quad (2)$$

where $\mathcal{T} = \text{Top-}k(\mathbf{r})$ denotes the set of selected experts.

### 3.2. The Non-linearity Mismatch in MoE Merging

Existing model merging methods, such as Task Arithmetic (Ilharco et al., 2023) and RegMean (Jin et al., 2025), typically rely on Linear Mode Connectivity (LMC) (Frankle et al., 2020), assuming that linear interpolation in parameter space approximates functional outcomes. While effective for dense architectures, this approach presumes a smooth, locally linear optimization landscape.

However, such assumptions break down in MoE architectures due to the non-linear routing mechanism. Consider a collection of task-specific MoE models $\{\mathcal{M}_i\}_{i=1}^{D}$, each with its own routing matrix $\mathbf{W}_i$. For a given input $\mathbf{x}$, the routing logit vector of model $\mathcal{M}_i$ is computed as $\mathbf{z}_i = \mathbf{W}_i \mathbf{x}$, producing the routing probability distribution $\mathbf{r}_i = \text{softmax}(\mathbf{z}_i)$.

Most linear merging approaches can be unified as a weighted aggregation in the parameter space: $\mathbf{W}_m = \sum_{i=1}^{D} \lambda_i \mathbf{W}_i$, where $\lambda_i$ is the merging coefficient for model $\mathcal{M}_i$. This linear interpolation naturally extends to the logits, yielding $\mathbf{z}_m = \mathbf{W}_m \mathbf{x} = \sum_{i=1}^{D} \lambda_i \mathbf{z}_i$. However, applying the softmax activation to these aggregated logits introduces a critical **non-linearity mismatch**. Due to the convexity of the log-sum-exp operation underlying softmax normalization, the routing distribution of the merged model, $\mathbf{r}_m$, is not equivalent to the aggregated routing distributions of the source models:

$$\mathbf{r}_m = \text{softmax}\left(\sum_{i=1}^{D} \lambda_i \mathbf{z}_i\right) \neq \sum_{i=1}^{D} \lambda_i \text{softmax}(\mathbf{z}_i). \quad (3)$$

This discrepancy is further amplified by the discrete Top-$k$ selection. Let $\mathcal{T}_i = \text{Top-}k(\mathbf{r}_i)$ and $\mathcal{T}_m = \text{Top-}k(\mathbf{r}_m)$ denote the expert sets selected by the source model $\mathcal{M}_i$ and the merged model, respectively. Because the softmax output is highly sensitive to logit perturbations, even minor discrepancies can alter the discrete ranking ($\mathcal{T}_m \neq \mathcal{T}_i$), causing *expert mismatch*. Consequently, the merged model may activate experts that are not optimized for the current input, leading to severely degraded representations.

These observations highlight that the central challenge lies in preserving routing behavior. Effective merging must explicitly account for the alignment of routing distributions, rather than relying solely on linear parameter aggregation.

## 4. Methodology

In this section, we introduce the **Hessian-Aware Router Calibration (HARC)** framework. We begin by formalizing the routing interference problem as a distribution alignment task. To overcome the intractability of the non-linear objective, we provide a rigorous theoretical analysis that establishes a quadratic approximation of the divergence via a

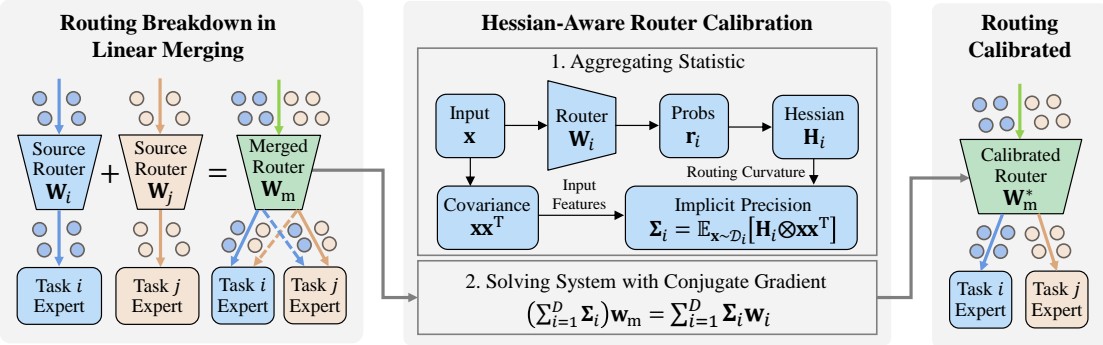

*Figure 2.* Overview of the Hessian-Aware Router Calibration (HARC) framework. **(Left)** Linear merging of source routers ($\mathbf{W}_i, \mathbf{W}_j$) disrupts the non-linear gating dynamics, leading to *routing breakdown* where tokens are misrouted to incorrect experts (dashed lines). **(Middle)** HARC addresses this by aggregating second-order statistics, the Hessian matrix $\mathbf{H}_i$ (routing curvature) and input covariance $\mathbf{x}\mathbf{x}^\top$, to form an implicit precision matrix $\mathbf{\Sigma}_i$ via Kronecker products. The optimal merged weights $\mathbf{w}_m$ are then solved efficiently using a matrix-free conjugate gradient algorithm. **(Right)** The resulting calibrated router restores correct expert assignments for each task.

second-order expansion. Building on this theoretical insight, we derive a closed-form solution for optimal router merging. Finally, to ensure scalability for large-scale models, we introduce a matrix-free conjugate gradient algorithm that efficiently solves the optimization problem without explicit matrix materialization.

### 4.1. Problem Formulation

The core challenge in MoE merging is to ensure that the merged router preserves the expert selection logic of the original task-specific routers. Naive averaging disrupts this logic due to the non-linearity of the softmax function. Therefore, our objective is to find a unified $\mathbf{W}_m$ that minimizes the discrepancy between the merged routing behavior and the individual task behaviors.

Let $\mathcal{D}_i$ denote the data distribution associated with task $i$. We formulate this objective by minimizing the expected Kullback-Leibler (KL) divergence over all tasks:

$$\mathbf{W}_m^* = \arg\min_{\mathbf{W}_m} \sum_{i=1}^{D} \mathbb{E}_{\mathbf{x}\sim\mathcal{D}_i}\left[\mathrm{KL}(\mathbf{r}_i(\mathbf{x}) \,\|\, \mathbf{r}_m(\mathbf{x}; \mathbf{W}_m))\right],$$

(4)

where $\mathbf{r}_m(\mathbf{x}; \mathbf{W}_m) = \mathrm{softmax}(\mathbf{W}_m\mathbf{x})$ denotes the routing distribution parameterized by $\mathbf{W}_m$. This objective encourages the merged router to match the full probability distribution of the source routers, rather than just their logits.

### 4.2. Theoretical Analysis

Directly optimizing the KL divergence is computationally intractable because the non-linearity of the softmax function prevents a simple analytical solution. To address this, we analyze the local geometry of the routing manifold using a second-order Taylor approximation.

**Lemma 4.1** (Quadratic Approximation of Routing Divergence)**.** *Let $\mathbf{r}_i$ and $\mathbf{r}_m$ be the probability distributions output*

*by the source and merged routers, respectively. Assuming that the merged logits $\mathbf{z}_m$ lie within the local neighborhood of the source logits $\mathbf{z}_i$, the KL divergence can be effectively approximated by a weighted quadratic form:*

$$\mathrm{KL}(\mathbf{r}_i\|\mathbf{r}_m) \approx \frac{1}{2}(\mathbf{z}_m - \mathbf{z}_i)^\top \mathbf{H}_i(\mathbf{z}_m - \mathbf{z}_i),$$

(5)

*where $\mathbf{H}_i = \mathrm{diag}(\mathbf{r}_i) - \mathbf{r}_i\mathbf{r}_i^\top$ is the Hessian matrix of the log-partition function.*

*Proof.* The proof follows from viewing the KL divergence as a Bregman divergence generated by the log-partition function $f(\mathbf{z}) = \log\sum_j e^{z^{(j)}}$. We perform a second-order Taylor expansion of $f(\mathbf{z}_m)$ around $\mathbf{z}_i$. Since the gradient is $\nabla f(\mathbf{z}_i) = \mathbf{r}_i$ and the Hessian is $\nabla^2 f(\mathbf{z}_i) = \mathbf{H}_i$, the first-order terms cancel out, leaving the quadratic term as the dominant component of the error. (See Appendix A.1 for the detailed derivation and validation of the local assumption).

**Interpretation.** Lemma 1 reveals that the Hessian $\mathbf{H}_i$ acts as a structure-aware metric. Its **diagonal terms**, $[\mathbf{H}_i]_{kk} = r_i^{(k)}(1-r_i^{(k)})$, function as an attention mechanism. Because MoE routers distribute probabilities across many experts under load-balancing constraints, individual probabilities typically fall well below $0.5$. In this operating regime, the diagonal weight increases monotonically with $r_i^{(k)}$, naturally assigning higher alignment penalties to the active experts while suppressing noise from inactive ones. Crucially, the **off-diagonal terms**, $[\mathbf{H}_i]_{kj} = -r_i^{(k)}r_i^{(j)}$, explicitly capture the competitive dynamics induced by the softmax constraint ($\sum r = 1$). By modeling this dense curvature, HARC preserves the precise relative ranking between experts, which diagonal approximations fail to capture.

Substituting the linear relationship $\mathbf{z} = \mathbf{W}\mathbf{x}$ into Lemma 1 allows us to reformulate the global objective into a solvable least-squares problem.

**Theorem 4.2** (Optimal Hessian-Aware Merging). *Let $\mathbf{w} = \text{vec}(\mathbf{W}^\top) \in \mathbb{R}^{Kd}$ be the vectorized form of the router parameters. We define the task-specific **precision matrix** $\boldsymbol{\Sigma}_i$ as the expectation of the Kronecker product between the routing Hessian and the input covariance:*

$$\boldsymbol{\Sigma}_i = \mathbb{E}_{\mathbf{x} \sim \mathcal{D}_i}[\mathbf{H}_i \otimes \mathbf{x}\mathbf{x}^\top]. \tag{6}$$

*The merged weights $\mathbf{w}_m^*$ that minimize the quadratic approximation of the total interference are given by the solution to the following linear system:*

$$\left( \sum_{i=1}^{D} \boldsymbol{\Sigma}_i \right) \mathbf{w}_m^* = \sum_{i=1}^{D} \boldsymbol{\Sigma}_i \mathbf{w}_i. \tag{7}$$

*Proof.* Let $\mathcal{J}(\mathbf{w}_m)$ denote the total objective function defined in Eq. 4. By substituting the quadratic approximation from Lemma 1 and utilizing the vectorization identity $(\mathbf{W}_m - \mathbf{W}_i)\mathbf{x} = (\mathbf{I}_K \otimes \mathbf{x}^\top)(\mathbf{w}_m - \mathbf{w}_i)$, the expected divergence $\mathcal{J}(\mathbf{w}_m)$ transforms into a weighted squared error:

$$\mathcal{J}(\mathbf{w}_m) \approx \sum_{i=1}^{D} \frac{1}{2}(\mathbf{w}_m - \mathbf{w}_i)^\top \boldsymbol{\Sigma}_i (\mathbf{w}_m - \mathbf{w}_i). \tag{8}$$

Setting the gradient $\nabla_{\mathbf{w}_m} \mathcal{J}$ to zero yields the normal equations presented in Eq. 7. The optimal solution $\mathbf{w}_m^* = (\sum \boldsymbol{\Sigma}_i)^{-1}(\sum \boldsymbol{\Sigma}_i \mathbf{w}_i)$ can be interpreted as a **Hessian-weighted average**, where parameter directions are weighted by their importance to the correct routing decisions (See Appendix A.2 for detailed derivation).

### 4.3. Efficient Computation via Matrix-Free Conjugate Gradient

While Theorem 1 provides a theoretically optimal closed-form solution, explicitly computing and inverting the aggregate precision matrix $\boldsymbol{\Sigma} = \sum \boldsymbol{\Sigma}_i$ is impractical. The matrix $\boldsymbol{\Sigma}$ has dimensions $Kd \times Kd$, leading to a prohibitive space complexity of $O(K^2 d^2)$. To make our method scalable to modern MoE architectures, we propose a matrix-free optimization strategy reinforced by diagonal regularization.

#### 4.3.1. DIAGONAL REGULARIZATION

In practice, the routing distribution is often sparse (due to Top-$k$ gating), causing the Hessian $\mathbf{H}_i$ to be rank-deficient. Direct inversion in such ill-conditioned settings leads to numerical instability. To address this, we introduce a regularization term incorporating a prior belief that parameter interactions are locally independent. This stabilizes the solution in directions where the Hessian has low curvature (i.e.,

high uncertainty). The regularized objective is given by:

$$\begin{aligned} \mathcal{J}_{\text{reg}}(\mathbf{w}_m) = \sum_{i=1}^{D} \frac{1}{2}(\mathbf{w}_m - \mathbf{w}_i)^\top \boldsymbol{\Sigma}_i (\mathbf{w}_m - \mathbf{w}_i) \\ + \gamma \sum_{i=1}^{D} (\mathbf{w}_m - \mathbf{w}_i)^\top \boldsymbol{\Lambda}_i (\mathbf{w}_m - \mathbf{w}_i), \end{aligned} \tag{9}$$

where $\boldsymbol{\Lambda}_i = \text{diag}(\boldsymbol{\Sigma}_i)$ extracts the diagonal elements of the precision matrix, and $\gamma > 0$ is a hyperparameter controlling the regularization strength.

The addition of $\gamma \boldsymbol{\Lambda}_i$ strengthens the diagonal dominance of the system. Following RegMean (Jin et al., 2025), we formulate the final precision matrix by interpolating between the full covariance and its diagonal approximation, rather than simply adding to the diagonal. The resulting regularized precision matrix $\boldsymbol{\Sigma}_{\text{reg},i}$ is defined as:

$$\boldsymbol{\Sigma}_{\text{reg},i} = \alpha \boldsymbol{\Sigma}_i + (1 - \alpha)\boldsymbol{\Lambda}_i, \tag{10}$$

where $\alpha \in [0, 1]$ is a damping factor derived from $\gamma$. When $\alpha = 1$, we use the full precision matrix; when $\alpha = 0$, the method reduces to a diagonal approximation (similar to Fisher Merging(Matena & Raffel, 2022)). This regularization leads to the final linear system solved by our algorithm:

$$\left( \sum_{i=1}^{D} \boldsymbol{\Sigma}_{\text{reg},i} \right) \mathbf{w}_m = \sum_{i=1}^{D} \boldsymbol{\Sigma}_{\text{reg},i} \mathbf{w}_i. \tag{11}$$

#### 4.3.2. MATRIX-FREE CONJUGATE GRADIENT

To avoid the prohibitive cost of explicitly constructing $\boldsymbol{\Sigma} = \sum_i \boldsymbol{\Sigma}_{\text{reg},i}$, we leverage the fact that the CG algorithm only requires matrix-vector products $\boldsymbol{\Sigma}_{\text{reg},i}\mathbf{w}$ rather than the full matrix. We therefore derive a *matrix-free* formulation to efficiently compute these products, enabling CG to solve Eq. 11 without explicitly constructing $\boldsymbol{\Sigma}_{\text{reg},i}$.

**Proposition 4.3** (Matrix-Free Linear Operator). *The matrix-vector product $\boldsymbol{\Sigma}_{reg,i}\mathbf{w}$ can be computed without materializing the full $Kd \times Kd$ matrix. Specifically, for a given input $\mathbf{x}$ and router weights $\mathbf{W}$ (where $\mathbf{w} = \text{vec}(\mathbf{W}^\top)$), we have:*

$$\begin{aligned} \boldsymbol{\Sigma}_{reg,i}\mathbf{w} = \mathbb{E}_{\mathbf{x} \sim \mathcal{D}_i} \big[ &\alpha \cdot \text{vec}\big(\mathbf{x}\left((\mathbf{W}\mathbf{x})^\top \mathbf{H}_i\right)\big) \\ &+ (1 - \alpha) \cdot \text{vec}\big(\text{diag}(\mathbf{x} \odot \mathbf{x}) \mathbf{W}^\top \text{diag}(\mathbf{H}_i)\big)\big] \end{aligned} \tag{12}$$

*where $\odot$ denotes element-wise multiplication.*

*Derivation.* By applying the Kronecker–vectorization identity, the dense term decouples $\mathbf{H}_i$ and $\mathbf{x}\mathbf{x}^\top$ into a sequence of efficient matrix–vector products via associativity. Meanwhile, the diagonal Kronecker property reduces the regularization term to element-wise scaling. (See Appendix A.3 for the complete derivation.)

---

**Algorithm 1** Hessian-Aware Router Calibration (HARC)

1: **Input:** Source models $\{\mathcal{M}_i\}_{i=1}^D$, uncalibrated merged model $\mathcal{M}_{\text{base}}$, calibration datasets $\{\mathcal{D}_i\}_{i=1}^D$, damping factor $\alpha$
2: **Output:** Calibrated model $\mathcal{M}_{\text{m}}$
3: **Initialize:** $\mathcal{M}_{\text{m}} \leftarrow \mathcal{M}_{\text{base}}$
4: **for** layer $l = 1$ to $L$ **do**
5:     Extract current router weights $\mathbf{W}_{\text{init}} \leftarrow \mathcal{M}_{\text{m}}^{(l)}$.
6:     Initialize the right-hand side (RHS) $\mathbf{b} \leftarrow \mathbf{0}$.
7:     **1. Aggregating Statistics**
8:     **for** $i = 1$ to $D$ **do**
9:         $\{\mathbf{x}_{i,j}\}_{j=1}^{N_i} \leftarrow \text{Forward}(\mathcal{M}_{\text{m}}[0\ldots l-1], \mathcal{D}_i)$
10:         **for** $j = 1$ to $N_i$ **do**
11:             Compute router output $\mathbf{r}_{i,j}$ and Hessian $\mathbf{H}_{i,j}$.
12:         **end for**
13:         Def operator $\mathcal{A}_i(\mathbf{w}) \triangleq \sum_{i=1}^D \boldsymbol{\Sigma}_{\text{reg},i}\mathbf{w}$ via Eq. 12.
14:         Accumulate $\mathbf{b} \leftarrow \mathbf{b} + \mathcal{A}_i(\mathbf{w}_i)$.
15:     **end for**
16:     **2. Solving System with Conjugate Gradient (CG)**
17:     Def operator $\mathcal{A}(\mathbf{w}) \triangleq \sum_{i=1}^D \mathcal{A}_i(\mathbf{w})$.
18:     $\mathbf{w}_{\text{m}}^* \leftarrow \text{CGSolver}(\mathcal{A}, \mathbf{b}, \text{vec}(\mathbf{W}_{\text{init}}^\top))$.
19:     // See Algorithm 2 in Appendix B.1 for details.
20:     **3. Updating Model**
21:     Update router weights in $\mathcal{M}_{\text{m}}^{(l)}$ with $\mathbf{w}_{\text{m}}^*$.
22: **end for**
23: **Return** $\mathcal{M}_{\text{m}}$

---

**Significance.** This decomposition reduces both the time and space complexity from quadratic $\mathcal{O}(K^2d^2)$ to linear $\mathcal{O}(Kd)$ with respect to the hidden dimension. It allows us to employ the CG algorithm to iteratively solve for $\mathbf{w}_{\text{m}}^*$ with high efficiency, scaling seamlessly to large models.

### 4.4. Algorithm

We present the router calibration procedure in Algorithm 1. Our method initializes from an uncalibrated merged model $\mathcal{M}_{\text{base}}$ and is agnostic to the specific model merging strategy (e.g., Task Arithmetic (Ilharco et al., 2023), TIES-Merging (Yadav et al., 2023)) employed. It progressively refines the routing parameters $\mathbf{W}_{\text{m}}^{(l)}$ for each MoE layer $l$. Crucially, the input features $\mathbf{x}$ used to calibrate layer $l$ are generated on the fly by forwarding the merged model up to layer $l-1$. This design ensures that each router is optimized with respect to the distribution shifts induced by parameter merging in preceding layers. The resulting optimization problem is solved using a matrix-free conjugate gradient method, exploiting the Kronecker-product structure in Section 4.3 to avoid explicit Hessian materialization. As detailed in Appendix C, replacing the explicit analytical calculation with our matrix-free formulation reduces the overall optimization time complexity across all $L$ layers from a prohibitive $\mathcal{O}(L(N_{\text{total}}K^2d^2 + K^3d^3))$ to $\mathcal{O}(L \cdot T \cdot N_{\text{total}} \cdot Kd)$, where

$N_{\text{total}}$ is the total number of calibration tokens and $T$ is the number of CG iterations. Correspondingly, the peak space complexity drops from $\mathcal{O}(K^2d^2)$ to $\mathcal{O}(N_{\text{total}}d + Kd)$.

### 4.5. Discussion

In this section, we analyze the relationship between HARC and existing data-driven merging paradigms. We highlight how our Hessian-aware formulation generalizes previous linear regression-based methods by capturing the non-linear routing dynamics identified in Lemma 1.

**Connection to Fisher Merging (Capturing Competitive Dynamics).** Fisher Merging (Matena & Raffel, 2022) weighs parameters based on the diagonal of the Fisher Information Matrix (FIM). Theoretically, our precision matrix $\boldsymbol{\Sigma}_i$ corresponds to the exact empirical FIM. However, standard Fisher Merging assumes a diagonal FIM to reduce memory costs. As discussed in our theoretical analysis, the MoE softmax constraint ($\sum r^{(k)} = 1$) introduces strong negative correlations between experts (the **off-diagonal competitive dynamics**). By assuming independence, diagonal Fisher Merging fails to model these trade-offs. HARC preserves these dense correlations via Kronecker factorization, effectively extending Fisher Merging to the non-linear routing context without prohibitive memory costs.

**Connection to RegMean (Enabling Importance Weighting).** RegMean (Jin et al., 2025) minimizes the $L_2$ distance between activations, which mathematically implies an identity Hessian assumption ($\mathbf{H}_i \approx \mathbf{I}$). This effectively treats all experts as equally important, regardless of their activation probability. In contrast, HARC incorporates the true routing curvature $\mathbf{H}_i$, which acts as an **importance weighting** mechanism (Lemma 1). This ensures that the merged router prioritizes alignment for *active experts* (high curvature regions) while tolerating errors in inactive ones. This distinction explains why HARC significantly outperforms RegMean in addressing the *expert mismatch* problem, as validated in our experiments.

## 5. Experiments

### 5.1. Experimental Setup

**Models and Dataset.** We validate our method on the MoE-based model OLMoE-1B-7B-0125 (Muennighoff et al., 2025). Domain-specific models are fine-tuned on subsets of mathematical reasoning and code generation from the OLMoE-SFT dataset. For methods requiring calibration data, we sample prompts from OpenMathInstruct2 (Toshniwal et al., 2024) and SelfOSSInstructSC2[1]. Responses are generated by the source model, retaining only those

---

[1] https://huggingface.co/datasets/bigcode/self-oss-instruct-sc2-instructions

*Table 1.* Multi-task performance when merging OLMoE models on mathematics reasoning and code generation tasks.

| Method | Math | | | Code | | | Overall |
|---|---|---|---|---|---|---|---|
| | GSM8K | MATH500 | Average | HumanEval+ | MBPP+ | Average | |
| Individual | 69.20 | 17.53 | 43.37 | 33.50 | 39.95 | 36.73 | 40.05 |
| Weight Averaging | 65.53 | 16.86 | 41.20 | 20.27 | 35.07 | 27.67 | 34.43 |
| w/ HARC | 65.54 | 16.64 | 41.09 | 20.73 | 36.82 | 28.78 | 34.93 |
| TIES-Merging | 66.40 | 15.75 | 41.08 | 26.45 | 35.93 | 31.19 | 36.13 |
| w/ HARC | 67.07 | 16.48 | 41.78 | 27.29 | 35.83 | 31.56 | 36.67 |
| DARE | 65.34 | 14.98 | 40.16 | 29.57 | 36.77 | 33.17 | 36.67 |
| w/ HARC | 65.58 | 15.05 | 40.32 | 31.36 | 36.82 | 34.09 | 37.20 |
| WUDI-Merging | 69.45 | 17.43 | 43.44 | 28.73 | 36.72 | 32.73 | 38.08 |
| w/ HARC | 70.06 | 17.96 | 44.01 | 29.61 | 38.19 | 33.90 | 38.96 |
| Fisher Merging | 67.23 | 18.35 | 42.79 | 19.70 | 34.85 | 27.28 | 35.03 |
| w/ HARC | 66.87 | 17.51 | 42.19 | 20.39 | 36.03 | 28.21 | 35.20 |
| RegMean | 70.48 | 16.80 | 43.64 | 24.47 | 36.81 | 30.64 | 37.14 |
| w/ HARC | 70.76 | 17.41 | 44.09 | 25.11 | 37.04 | 31.08 | 37.58 |

matching ground truth (math) or passing unit tests (code). Additional experiments on merging three models and on the larger Qwen3-30B-A3B (Yang et al., 2025) are presented in Appendices D.1 and D.2, respectively.

**Baselines.** We evaluate HARC against representative data-free merging methods, including Weight Averaging (Wortsman et al., 2022), TIES-Merging (Yadav et al., 2023), DARE-TIES (Yu et al., 2024), and WUDI-Merging (Cheng et al., 2025), as well as data-driven approaches such as Fisher Merging (Matena & Raffel, 2022) and RegMean (Jin et al., 2025). To establish an empirical upper bound, we also report *Individual* results, corresponding to the performance of each source model on its respective domain.

**Evaluation.** We evaluate the performance of the above models on math benchmarks (GSM8K (Cobbe et al., 2021) and MATH500 (Lin et al., 2025)) and code benchmarks (HumanEval+ and MBPP+ (Liu et al., 2023)). To mitigate evaluation variance, we sample 16 outputs per problem at temperature 0.1, and report the average accuracy.

### 5.2. Main Results

Table 1 reports the results of merging two OLMoE models fine-tuned for mathematics and code generation, respectively. Overall, augmenting existing merging pipelines with our Hessian-Aware Router Calibration (HARC) consistently improves multi-task performance.

**Consistent Improvement Across Merging Strategies.** HARC yields improvements when applied to a diverse set of baselines, ranging from simple Weight Averaging to the stronger WUDI framework. For example, under WUDI, HARC improves the overall score from 38.08 to **38.96** (+0.88), establishing the best performance among all merged variants in Table 1. This pattern indicates that routing calibration is complementary to how expert parameters

are merged, and it can be plugged into different merging strategies as a lightweight post-processing step.

**Mitigating Routing Collapse in Sensitive Domains.** A second observation is that naive merging can severely damage code generation performance, which is typically more sensitive to expert selection. For instance, Average-Merging substantially degrades the code score relative to the individual model (e.g., HumanEval+ drops from 33.50 to 20.27). HARC partially recovers this loss by correcting routing behavior. In the WUDI setting, HARC increases MBPP+ from 36.72 to 38.19 and improves the code average from 32.73 to 33.90. These results are consistent with our analysis in Section 3: because the router involves a non-linear softmax and discrete Top-$k$ selection, small logit perturbations can flip expert assignments. Aligning the router distribution, rather than only averaging parameters, is therefore crucial for preserving domain-specific capabilities.

### 5.3. Ablation Studies

**Effectiveness of Hessian Information.** To quantify the contribution of second-order curvature, we compare our full Hessian formulation with three simplified variants: (i) a diagonal Hessian approximation, (ii) an identity approximation ($\mathbf{H}_i = \mathbf{I}$), and (iii) the uncalibrated baseline (i.e., without router calibration). As shown in Figure 3(a), performance degrades monotonically as the curvature structure is simplified, indicating that richer curvature information leads to more accurate routing alignment. Notably, the gap between the full and diagonal settings (with a drop of 0.12) suggests that off-diagonal terms are not negligible. This is expected because the softmax normalization couples expert logits, inducing correlated perturbations across routing dimensions. Moreover, Hessian-based variants consistently outperform the identity approximation, showing that aligning router behavior cannot be reduced to input-statistics

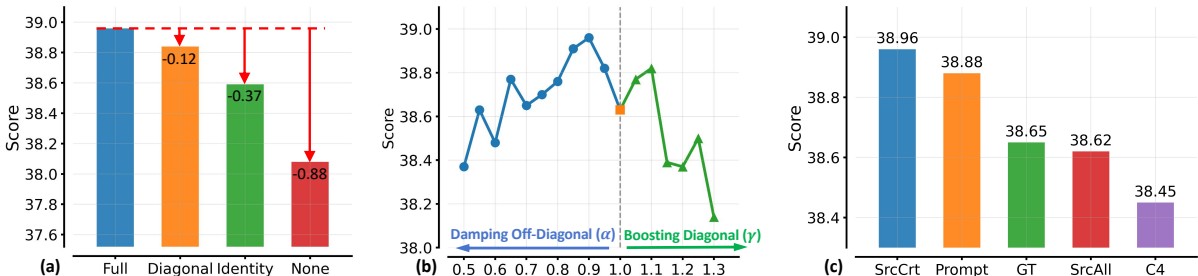

*Figure 3.* **(a)** Effectiveness of Hessian information: Comparison of various Hessian structures in WUDI-Merging. *Full* uses the complete Hessian matrix; *Diagonal* retains only the diagonal elements; *Identity* assumes an identity Hessian; and *None* represents the uncalibrated baseline. **(b)** Effectiveness of diagonal regularization: Comparison of the impact of reducing off-diagonal elements via the damping factor $\alpha$ (left) versus increasing diagonal elements via the scaling factor $\gamma$ (right). **(c)** Impact of data composition: Comparison of different types of calibration data. *SrcCrt* uses filtered correct source responses; *Prompt* uses only the prompts; *GT* uses ground truth references; *SrcAll* includes unfiltered source responses with potential errors; and *C4* uses general-purpose pre-training data (C4-en).

matching alone. Incorporating curvature allows the calibration to emphasize high-sensitivity regions of the routing function, which yields substantial gains of up to $0.88$ points over the uncalibrated baseline.

**Effectiveness of Diagonal Regularization.** We study the impact of regularization by varying the damping factor $\alpha$ (off-diagonal suppression) and the diagonal scaling factor $\gamma$ (diagonal amplification), as shown in Figure 3(b). Although both approaches increase diagonal dominance in principle, they lead to different empirical stability profiles. Performance deteriorates when $\alpha$ is too small or $\gamma$ is too large, indicating that overly aggressive regularization distorts routing decisions. Across a broad range of values, the $\alpha$-based strategy remains stable and consistently improves over the baseline. In contrast, the $\gamma$-based strategy degrades sharply once $1 + \gamma$ exceeds $1.15$. We attribute this to numerical effects during implementation. Our solver accumulates precision statistics over many batches, and explicitly inflating diagonal magnitudes amplifies floating-point errors during aggregation. Suppressing off-diagonal entries via $\alpha$ avoids scale inflation, which better preserves numerical precision and leads to more reliable convergence. Notably, the fixed $\alpha = 0.9$ serves as a robust default: it is used across all six merging baselines in Table 1 without any per-method tuning.

### 5.4. Further Analysis

**Impact of Data Composition.** We investigate how data composition affects alignment by comparing filtered correct source generations (`SrcCrt`), unfiltered generations (`SrcAll`), ground truth targets (`GT`), prompts only (`Prompt`), and general-purpose pre-training data (`C4`). As shown in Figure 3(c), `SrcCrt` achieves the highest performance, confirming that the optimal calibration target is the model's native "on-policy" distribution corresponding to correct reasoning. Surprisingly, the `Prompt` setting yields comparable results, outperforming strategies that utilize full response text like `GT` and `SrcAll`. This implies

that the initial routing decisions triggered by input instructions are paramount; correctly dispatching the prompt to domain-relevant experts establishes a robust routing trajectory that persists during generation. In contrast, `GT` suffers from distribution mismatch—forcing alignment with text the model did not generate pushes the router into incompatible states—while `SrcAll` degrades performance by incorporating erroneous reasoning traces into the Hessian estimation. Notably, we also evaluate with C4-en (Raffel et al., 2020), a general-purpose pre-training corpus unrelated to the downstream tasks. HARC still retains $42\%$ of the gain ($+0.37$ vs. $+0.88$), confirming that while high-quality data is beneficial, HARC provides consistent gains across a wide spectrum of data availability and quality.

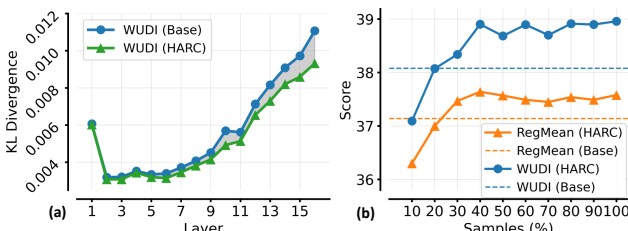

*Figure 4.* Routing consistency and data efficiency analysis. (a) Layer-wise KL divergence between merged and source routers. (b) Performance trajectory with varying amounts of calibration data.

**Mitigating Cumulative Routing Deviation.** To assess whether HARC effectively mitigates the routing breakdown phenomenon, we visualize the layer-wise Kullback-Leibler (KL) divergence between the merged router's output distribution and the source routers' distributions in Figure 4(a). We observe that, for the baseline WUDI method, the divergence remains low in the shallower layers but experiences a sharp exponential increase starting from Layer 11. This supports our hypothesis that routing errors accumulate as network depth increases, likely due to the increasing specialization of experts in the deeper layers. In contrast, applying HARC suppresses this divergence trend. By incorporating Hessian curvature information, HARC maintains a much closer alignment with the original routing behavior, partic-

ularly in the critical deeper layers (layers 12-16), thereby effectively preventing the catastrophic semantic drift caused by expert mismatch during model merging.

**Data Efficiency.** Given that HARC relies on calibration data to estimate Hessian and precision matrices, we explore the sensitivity of our method to the size of the calibration set. Figure 4(b) shows the performance of RegMean and WUDI-Merging augmented with HARC as the percentage of calibration samples increases from 10% to 100%. The results demonstrate that HARC is highly data-efficient. Both methods exceed their respective uncalibrated baselines (indicated by dashed lines) using only 20% of the available calibration data. Furthermore, performance stabilizes rapidly, reaching a plateau around 40% of the samples. This indicates that the estimated second-order statistics converge quickly to a sufficient precision for effective alignment, making HARC a practical solution even in scenarios with limited data or strict computational constraints.

## 6. Conclusion

In this work, we identify *routing breakdown* as a critical failure mode in MoE merging, stemming from the high sensitivity of non-linear gating mechanisms to parameter perturbations. To address this, we propose Hessian-Aware Router Calibration (HARC), a training-free framework that leverages second-order curvature information to analytically realign the merged router via a scalable, matrix-free conjugate gradient solver. Experiments across math and code generation tasks demonstrate that HARC effectively rectifies expert mismatch and consistently outperforms existing merging baselines. Our findings highlight that preserving routing logic is as essential as aligning parameter weights in sparse architectures, offering a robust and efficient path for consolidating MoE models without the need for retraining.

## Acknowledgements

This work was supported by the National Natural Science Foundation of China (No. 62576368).

## Use of Generative AI Tools

We used large language models as a writing assistant to refine phrasing and improve the readability of the manuscript. All technical content, claims, experimental results, and conclusions were produced, verified, and validated by the authors, who take full responsibility for the final manuscript.

## Impact Statement

This work advances model merging techniques for Mixture-of-Experts (MoE) language models by identifying and addressing routing breakdown, a key failure mode that can undermine the safe and effective reuse of pretrained models. By enabling reliable consolidation of multiple expert models without retraining, the proposed Hessian-Aware Router Calibration (HARC) method can reduce computational costs, energy consumption, and barriers to deploying capable models, with positive environmental and accessibility implications. At the same time, improved model merging may amplify the capabilities of existing systems, underscoring the importance of responsible deployment and evaluation, particularly in high-stakes applications. As this work focuses on training-free calibration rather than model design or data collection, ethical risks primarily relate to downstream use; these should be managed through appropriate oversight, benchmarking, and domain-specific safeguards.

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

# A. Proofs and Assumption Validation

In this section, we provide detailed proofs for the theoretical results presented in Section 4 and validate the key assumption underlying Lemma 4.1.

## A.1. Proof and Assumption Validation of Lemma 4.1

*Proof.* Let the routing probability be defined as $\mathbf{r}(\mathbf{z}) = \mathrm{softmax}(\mathbf{z})$. The KL divergence is given by:

$$\mathrm{KL}(\mathbf{r}_i \| \mathbf{r}_\mathrm{m}) = \sum_{k=1}^{K} r_i^{(k)} \log \frac{r_i^{(k)}}{r_\mathrm{m}^{(k)}}. \tag{13}$$

Let $f(\mathbf{z}) = \log \sum_j e^{z^{(j)}}$ be the log-partition function. We can rewrite the divergence in terms of the logits $\mathbf{z}$:

$$\mathrm{KL}(\mathbf{r}_i \| \mathbf{r}_\mathrm{m}) = \mathbf{r}_i^\top (\mathbf{z}_i - \mathbf{z}_\mathrm{m}) - f(\mathbf{z}_i) + f(\mathbf{z}_\mathrm{m}). \tag{14}$$

We apply a second-order Taylor expansion to $f(\mathbf{z}_\mathrm{m})$ around the point $\mathbf{z}_i$:

$$f(\mathbf{z}_\mathrm{m}) \approx f(\mathbf{z}_i) + \nabla f(\mathbf{z}_i)^\top (\mathbf{z}_\mathrm{m} - \mathbf{z}_i) + \frac{1}{2}(\mathbf{z}_\mathrm{m} - \mathbf{z}_i)^\top \nabla^2 f(\mathbf{z}_i)(\mathbf{z}_\mathrm{m} - \mathbf{z}_i). \tag{15}$$

We know that the gradient of the log-partition function is the probability vector itself, $\nabla f(\mathbf{z}_i) = \mathbf{r}_i$, and its Hessian is $\nabla^2 f(\mathbf{z}_i) = \mathrm{diag}(\mathbf{r}_i) - \mathbf{r}_i \mathbf{r}_i^\top = \mathbf{H}_i$. Substituting these into the expansion:

$$\begin{aligned}
\mathrm{KL}(\mathbf{r}_i \| \mathbf{r}_\mathrm{m}) &\approx \mathbf{r}_i^\top (\mathbf{z}_i - \mathbf{z}_\mathrm{m}) - f(\mathbf{z}_i) \\
&\quad + \left[ f(\mathbf{z}_i) + \mathbf{r}_i^\top (\mathbf{z}_\mathrm{m} - \mathbf{z}_i) + \frac{1}{2}(\mathbf{z}_\mathrm{m} - \mathbf{z}_i)^\top \mathbf{H}_i (\mathbf{z}_\mathrm{m} - \mathbf{z}_i) \right] \\
&= \frac{1}{2}(\mathbf{z}_\mathrm{m} - \mathbf{z}_i)^\top \mathbf{H}_i (\mathbf{z}_\mathrm{m} - \mathbf{z}_i).
\end{aligned} \tag{16}$$

This completes the proof. $\square$

**Assumption Validation.** The quadratic approximation assumes that the merged logits $\mathbf{z}_\mathrm{m}$ lie within the local neighborhood of the source logits $\mathbf{z}_i$. We validate this by measuring the relative logit shift $\|\mathbf{z}_\mathrm{m} - \mathbf{z}_i\| / \|\mathbf{z}_i\|$ on calibration data. Across TIES-Merging and WUDI-Merging, over 90% of tokens exhibit a shift below 0.1, and over 99% below 0.2. This is expected because the merging process itself constrains logit deviations: arithmetic-based methods produce deviations determined by task vector norms (typically small in fine-tuning), while optimization-based methods either explicitly minimize $\|\mathbf{z}_\mathrm{m} - \mathbf{z}_i\|$ or implicitly constrain it during interference reduction.

## A.2. Proof of Theorem 4.2

*Proof.* Based on Lemma 4.1, the total objective function is:

$$\mathcal{J}(\mathbf{W}_\mathrm{m}) \approx \sum_{i=1}^{D} \mathbb{E}_{\mathbf{x} \sim \mathcal{D}_i} \left[ \frac{1}{2}(\mathbf{z}_\mathrm{m} - \mathbf{z}_i)^\top \mathbf{H}_i (\mathbf{z}_\mathrm{m} - \mathbf{z}_i) \right]. \tag{17}$$

Substituting $\mathbf{z} = \mathbf{W}\mathbf{x}$ and vectorizing $\mathbf{w} = \mathrm{vec}(\mathbf{W}^\top)$:

$$\mathbf{z}_\mathrm{m} - \mathbf{z}_i = (\mathbf{W}_\mathrm{m} - \mathbf{W}_i)\mathbf{x} = (\mathbf{I}_K \otimes \mathbf{x}^\top)(\mathbf{w}_\mathrm{m} - \mathbf{w}_i). \tag{18}$$

The quadratic term becomes:

$$\begin{aligned}
(\mathbf{z}_\mathrm{m} - \mathbf{z}_i)^\top \mathbf{H}_i (\mathbf{z}_\mathrm{m} - \mathbf{z}_i) &= (\mathbf{w}_\mathrm{m} - \mathbf{w}_i)^\top (\mathbf{I} \otimes \mathbf{x}) \mathbf{H}_i (\mathbf{I} \otimes \mathbf{x}^\top)(\mathbf{w}_\mathrm{m} - \mathbf{w}_i) \\
&= (\mathbf{w}_\mathrm{m} - \mathbf{w}_i)^\top (\mathbf{H}_i \otimes \mathbf{x}\mathbf{x}^\top)(\mathbf{w}_\mathrm{m} - \mathbf{w}_i).
\end{aligned} \tag{19}$$

Taking the expectation over $\mathcal{D}_i$, we substitute $\boldsymbol{\Sigma}_i = \mathbb{E}[\mathbf{H}_i \otimes \mathbf{x}\mathbf{x}^\top]$:

$$\mathcal{J}(\mathbf{w}_\mathrm{m}) \approx \sum_{i=1}^{D} \frac{1}{2}(\mathbf{w}_\mathrm{m} - \mathbf{w}_i)^\top \boldsymbol{\Sigma}_i (\mathbf{w}_\mathrm{m} - \mathbf{w}_i). \tag{20}$$

Taking the gradient with respect to $\mathbf{w}_\mathrm{m}$ and setting it to zero:

$$\nabla\mathcal{J} = \sum_{i=1}^{D} \boldsymbol{\Sigma}_i(\mathbf{w}_\mathrm{m} - \mathbf{w}_i) = 0 \implies \left(\sum_{i=1}^{D}\boldsymbol{\Sigma}_i\right)\mathbf{w}_\mathrm{m} = \sum_{i=1}^{D}\boldsymbol{\Sigma}_i\mathbf{w}_i. \tag{21}$$

Assuming $\sum \boldsymbol{\Sigma}_i$ is invertible (or regularized), we obtain the closed-form solution. $\qquad \square$

### A.3. Derivation of Proposition 4.3

*Derivation.* By the definition of the regularized precision matrix (Eq. 10), we can explicitly expand the matrix-vector product $\boldsymbol{\Sigma}_{\mathrm{reg},i}\mathbf{w}$ into a dense component and a diagonal regularization component:

$$\boldsymbol{\Sigma}_{\mathrm{reg},i}\mathbf{w} = \mathbb{E}_{\mathbf{x}\sim\mathcal{D}_i}\left[\alpha(\mathbf{H}_i \otimes \mathbf{x}\mathbf{x}^\top)\mathbf{w} + (1-\alpha)\,\mathrm{diag}(\mathbf{H}_i \otimes \mathbf{x}\mathbf{x}^\top)\mathbf{w}\right]. \tag{22}$$

We now derive the matrix-free formulation for each term inside the expectation by substituting $\mathbf{w} = \mathrm{vec}(\mathbf{W}^\top)$.

**Dense term.** For the dense component $\alpha\boldsymbol{\Sigma}_i\mathbf{w}$, applying the standard identity $(\mathbf{A}\otimes\mathbf{B})\,\mathrm{vec}(\mathbf{Y}) = \mathrm{vec}(\mathbf{B}\mathbf{Y}\mathbf{A}^\top)$:

$$\begin{aligned}(\mathbf{H}_i \otimes \mathbf{x}\mathbf{x}^\top)\,\mathrm{vec}(\mathbf{W}^\top) &= \mathrm{vec}\big(\mathbf{x}\mathbf{x}^\top\mathbf{W}^\top\mathbf{H}_i\big) \\ &= \mathrm{vec}\big(\mathbf{x}\left((\mathbf{W}\mathbf{x})^\top\mathbf{H}_i\right)\big).\end{aligned} \tag{23}$$

This grouping explicitly reveals the matrix-free evaluation sequence: (1) compute logits $\mathbf{z} = \mathbf{W}\mathbf{x} \in \mathbb{R}^K$, (2) compute the Hessian-weighted row vector $\mathbf{u}^\top = \mathbf{z}^\top\mathbf{H}_i \in \mathbb{R}^{1\times K}$, and (3) form the outer product $\mathbf{x}\mathbf{u}^\top \in \mathbb{R}^{d\times K}$. Each step involves only matrix-vector or vector-matrix products, strictly avoiding any matrix-matrix multiplication. Specifically, step (1) takes $\mathcal{O}(Kd)$ time, step (2) takes $\mathcal{O}(K^2)$ time, and step (3) takes $\mathcal{O}(Kd)$ time. Given that the hidden dimension is typically much larger than the number of experts ($d \gg K$) in MoE architectures, the overall time complexity simplifies to $\mathcal{O}(Kd)$. Furthermore, the space complexity for this term is also bounded by $\mathcal{O}(Kd)$ to store the intermediate vectors, the routing Hessian, and the output matrix.

**Diagonal regularization term.** For the diagonal component, we utilize the property $\mathrm{diag}(\mathbf{A}\otimes\mathbf{B}) = \mathrm{diag}(\mathbf{A})\otimes\mathrm{diag}(\mathbf{B})$ and note that $\mathrm{diag}(\mathbf{x}\mathbf{x}^\top) = \mathrm{diag}(\mathbf{x}\odot\mathbf{x})$. Applying the standard vectorization identity to these diagonal factors yields:

$$\begin{aligned}\mathrm{diag}\left(\mathbf{H}_i \otimes \mathbf{x}\mathbf{x}^\top\right)\mathrm{vec}\left(\mathbf{W}^\top\right) &= \left(\mathrm{diag}\left(\mathbf{H}_i\right) \otimes \mathrm{diag}\left(\mathbf{x}\odot\mathbf{x}\right)\right)\mathrm{vec}\left(\mathbf{W}^\top\right) \\ &= \mathrm{vec}\left(\mathrm{diag}\left(\mathbf{x}\odot\mathbf{x}\right)\mathbf{W}^\top\,\mathrm{diag}\left(\mathbf{H}_i\right)\right).\end{aligned} \tag{24}$$

Element-wise, this corresponds to scaling each entry of $\mathbf{W}^\top$ by the respective diagonal factors: $[\cdot]_{jk} = x_j^2\,\mathbf{W}_{kj}\,[\mathbf{H}_i]_{kk}$, which constitutes a pure element-wise scaling operation requiring only $O(Kd)$ FLOPs. The space complexity is also strictly bounded by $O(Kd)$ to allocate the scaled matrix, plus $O(d)$ and $O(K)$ for the diagonal vectors.

**Combined operator.** Combining both terms and taking the expectation yields the exact matrix-free linear operator in Eq. 12. This formulation is mathematically rigorous under standard vectorization conventions and completely avoids materializing the full $Kd \times Kd$ precision matrix. Overall, our matrix-free formulation evaluates the combined operator in only $\mathcal{O}(Kd)$ time and space per instance, achieving an order-of-magnitude reduction in computational and memory overhead compared to the prohibitive $\mathcal{O}(K^2d^2)$ complexity of explicit materialization. $\qquad \square$

## B. Implementation Details

### B.1. Matrix-Free Conjugate Gradient Solver

We provide the detailed procedure of the Matrix-Free Conjugate Gradient solver used in HARC. This solver optimizes the quadratic objective without explicitly materializing the Precision matrix, relying solely on the matrix-vector product operator $\mathcal{A}(\cdot)$.

---

**Algorithm 2** Matrix-Free Conjugate Gradient Solver

---

1: **Input:** Linear operator $\mathcal{A}(\cdot)$, RHS vector $\mathbf{b}$, initial guess $\mathbf{w}_{\text{init}}$, tolerance $\epsilon$
2: **Output:** Optimized weights $\mathbf{w}^*$
3: **Initialize:**
4: Set initial guess $\mathbf{w} \leftarrow \mathbf{w}_{\text{init}}$
5: Compute initial residual $\mathbf{r} \leftarrow \mathbf{b} - \mathcal{A}(\mathbf{w})$
6: Set initial search direction $\mathbf{p} \leftarrow \mathbf{r}$
7: Compute initial squared residual norm $\rho_{\text{old}} \leftarrow \mathbf{r}^\top \mathbf{r}$
8: Compute target norm $b_{\text{norm}} \leftarrow \|\mathbf{b}\|_2$
9: **while** $\|\mathbf{r}\|_2 / b_{\text{norm}} > \epsilon$ **do**
10:    Compute matrix-free product $\mathbf{q} \leftarrow \mathcal{A}(\mathbf{p})$ via Eq. 12
11:    Calculate curvature $\eta \leftarrow \mathbf{p}^\top \mathbf{q}$
12:    Calculate optimal step size $\alpha_{\text{step}} \leftarrow \rho_{\text{old}}/\eta$
13:    Update solution weights $\mathbf{w} \leftarrow \mathbf{w} + \alpha_{\text{step}}\mathbf{p}$
14:    Update residual $\mathbf{r} \leftarrow \mathbf{r} - \alpha_{\text{step}}\mathbf{q}$
15:    Compute new squared residual norm $\rho_{\text{new}} \leftarrow \mathbf{r}^\top \mathbf{r}$
16:    Compute Gram-Schmidt coefficient $\beta \leftarrow \rho_{\text{new}}/\rho_{\text{old}}$
17:    Update search direction $\mathbf{p} \leftarrow \mathbf{r} + \beta\mathbf{p}$
18:    Update squared norm $\rho_{\text{old}} \leftarrow \rho_{\text{new}}$
19: **end while**
20: **Return** $\mathbf{w}$

---

**Convergence Stability.** A potential concern is whether the CG solver converges when the routing Hessian $\mathbf{H}_i$ is highly sparse or specialized (e.g., with near-zero eigenvalues for inactive experts). Our diagonal regularization (Eq. 10) directly addresses this by interpolating the full precision matrix with its diagonal approximation, which lifts near-zero eigenvalues and reduces the condition number of the aggregated system. In practice, we observe stable convergence across all experimental settings: with a fixed tolerance of $10^{-6}$, the average iteration count is 256.1 for two-model merging and actually *decreases* to 231.9 for three-model merging. This acceleration arises because the precision matrices $\mathbf{H}_i$ from different domains have complementary null spaces: the positive eigenvalues of $\mathbf{H}_3$ fill the near-zero eigenvalues of $\mathbf{H}_1$ and $\mathbf{H}_2$, raising the minimum eigenvalue of the aggregate Hessian $\mathbf{H} = \sum_i \mathbf{H}_i$ and further reducing its condition number.

### B.2. Hyperparameters

For the HARC framework, we set the regularization damping factor to $\alpha = 0.9$. We solve the optimization using a matrix-free conjugate gradient algorithm with a convergence tolerance of $1 \times 10^{-6}$. For fair comparison, we tune all baseline hyperparameters (e.g., sparsification rates and scaling factors) via grid search on a validation set and report their best-performing results. Hyperparameters adopted for baseline methods can be found in Table 2.

*Table 2.* Hyperparameters for baseline methods.

| Method | Hyperparameters |
|---|---|
| TIES-Merging | Density: $0.6$, model weights: $(0.4, 0.6)$ |
| DARE | Density: $0.4$, model weights: $(0.3, 0.7)$ |
| WUDI-Merging | Optimizer: Adam, learning rate: $10^{-5}$, iteration steps: $300$ |
| Fisher Merging | Model weights: $(0.5, 0.5)$ |
| RegMean | Regularization factor: $0.9$ |

## C. Complexity and Empirical Overhead Analysis

In this section, we provide a theoretical complexity analysis of the complete HARC algorithm, followed by an empirical evaluation of its computational overhead and a discussion on the time-performance trade-off.

## C.1. Theoretical Complexity

To highlight the efficiency of our proposed solver, we compare the theoretical complexity of directly computing the closed-form solution versus our matrix-free conjugate gradient approach. Let $N_{\text{total}} = \sum_{i=1}^{D} N_i$ be the total number of calibration tokens, $L$ be the number of MoE layers, and $T$ be the average number of CG iterations.

**Direct Analytical Solution.** Directly computing $\mathbf{w}_{\text{m}}^* = (\sum \mathbf{\Sigma}_i)^{-1}(\sum \mathbf{\Sigma}_i \mathbf{w}_i)$ requires explicitly constructing and inverting the aggregated precision matrix $\mathbf{\Sigma}$, which has dimensions $Kd \times Kd$.

- *Space Complexity:* Materializing this full matrix demands $\mathcal{O}(K^2 d^2)$ memory per layer. For a modern MoE model (e.g., $d = 4096$, $K = 64$), storing just a single precision matrix in FP32 format would require over 250 GB of memory, instantly exceeding standard single-GPU capacities.

- *Time Complexity:* Constructing the matrix via full Kronecker products across all tokens takes $\mathcal{O}(N_{\text{total}} K^2 d^2)$ time. Furthermore, matrix inversion scales cubically, demanding $\mathcal{O}(K^3 d^3)$ time. The total time for calibrating $L$ layers is a prohibitive $\mathcal{O}(L \cdot (N_{\text{total}} K^2 d^2 + K^3 d^3))$.

**Matrix-Free CG Formulation.** Our approach entirely bypasses the explicit construction of $\mathbf{\Sigma}$ by leveraging the operator factorization derived in Appendix A.3, assuming $d \gg K$.

- *Space Complexity:* Calibrating sequentially layer-by-layer, the peak memory footprint is strictly dominated by caching the input features $\mathbf{x}$ and routing Hessians $\mathbf{H}$ for the current layer, requiring $\mathcal{O}(N_{\text{total}} \cdot d + N_{\text{total}} \cdot K^2)$ memory. Maintaining the CG state vectors (e.g., $\mathbf{w}, \mathbf{r}, \mathbf{p}, \mathbf{q}$) takes $\mathcal{O}(Kd)$. The overall space complexity reduces to $\mathcal{O}(N_{\text{total}} \cdot d + Kd)$.

- *Time Complexity:* The CG solver computes matrix-vector products on the fly. Evaluating the operator for all tokens in a single iteration takes $\mathcal{O}(N_{\text{total}} \cdot Kd)$ time. Thus, the optimization time for all layers is $\mathcal{O}(L \cdot T \cdot N_{\text{total}} \cdot Kd)$. Note that generating the calibration features via a forward pass takes an additional $\mathcal{O}(N_{\text{total}} \cdot \text{ForwardCost})$.

**Summary.** By replacing the cubic $\mathcal{O}(K^3 d^3)$ inversion and quadratic $\mathcal{O}(K^2 d^2)$ storage with linear operations bounded by $\mathcal{O}(Kd)$, HARC achieves an immense reduction in both computational and memory overhead. This structural optimization ensures that the calibration framework scales seamlessly to massive architectures.

## C.2. Empirical Overhead

We evaluate the actual computational cost of HARC compared to representative merging methods.

*Table 3.* Runtime and peak memory comparison across representative merging methods.

| Method | Runtime | Peak Memory |
|---|---|---|
| Data-free (Weight Averaging, TIES-Merging, DARE) | <5 min | ~80 GB |
| RegMean | 12.4 min | 120.9 GB |
| HARC | 36.2 min | 118.7 GB |

As shown in Table 3, HARC requires 36.2 minutes for full calibration across all 16 MoE layers. While this is longer than data-free methods and RegMean, HARC's peak memory footprint (118.7 GB) is actually lower than that of RegMean (120.9 GB). This empirical result aligns with our theoretical analysis: by leveraging the matrix-free formulation, HARC completely bypasses the massive memory spikes that arise when computing the closed-form solution directly.

## C.3. Time-Performance Trade-off

To explore the trade-off between calibration time and task performance, we evaluate a partial-layer calibration strategy, applying HARC only to the last $L$ MoE layers.

Table 4 reveals a flexible trade-off. Calibrating only the last 4 layers captures 36% of the full gain in just ~9 minutes, while the last 8 layers achieve 52% in ~18 minutes. This is consistent with our observation in Section 5.4 that routing deviation accumulates primarily in deeper layers. Users can thus adjust the number of calibrated layers to allocate their compute budget proportionally to their performance requirements.

*Table 4.* Partial-layer calibration: performance vs. runtime when applying HARC to the last $L$ layers (WUDI baseline).

| Layers | Runtime | Performance | % of Full Gain |
|---|---|---|---|
| 0 (baseline) | — | 38.08 | 0% |
| Last 4 | ∼9 min | 38.40 | 36% |
| Last 8 | ∼18 min | 38.54 | 52% |
| All 16 | ∼36 min | 38.96 | 100% |

# D. Extended Experimental Results

## D.1. Multi-Model Merging Scalability

To evaluate whether HARC scales beyond two-model merging, we extend our setup by adding a third OLMoE model fine-tuned for chat capabilities with training data derived from the OLMoE-SFT dataset. For the routing calibration of this chat task, we use prompts sampled from UltraFeedback (Cui et al., 2024). We then merge all three models and evaluate on chat benchmarks (IFEval (Zhou et al., 2023) and CommonsenseQA (Talmor et al., 2019)), math benchmarks (GSM8K (Cobbe et al., 2021) and MATH500 (Lin et al., 2025)), and code benchmarks (HumanEval+ and MBPP+ (Liu et al., 2023)).

*Table 5.* Multi-task performance when merging three OLMoE models (chat, math, and code).

| Method | Chat | | | Math | | | Code | | | Overall |
|---|---|---|---|---|---|---|---|---|---|---|
| | IFEval | CommQA | Average | GSM8K | MATH500 | Average | HumanEval+ | MBPP+ | Average | |
| Individual | 63.05 | 58.98 | 61.02 | 69.20 | 17.53 | 43.37 | 33.50 | 39.95 | 36.73 | 47.04 |
| Weight Averaging | 45.74 | 59.80 | 52.77 | 66.49 | 16.20 | 41.34 | 18.14 | 34.67 | 26.41 | 40.17 |
| w/ HARC | 46.21 | 59.50 | 52.85 | 67.34 | 16.79 | 42.06 | 18.96 | 34.83 | 26.90 | 40.60 |
| TIES-Merging | 48.90 | 61.54 | 55.22 | 66.56 | 15.91 | 41.23 | 19.70 | 33.76 | 26.73 | 41.06 |
| w/ HARC | 49.27 | 61.29 | 55.28 | 66.02 | 15.83 | 40.92 | 20.13 | 35.47 | 27.80 | 41.33 |
| DARE | 48.71 | 52.38 | 50.54 | 64.66 | 16.31 | 40.49 | 29.04 | 36.33 | 32.68 | 41.24 |
| w/ HARC | 49.03 | 53.24 | 51.13 | 65.37 | 16.81 | 41.09 | 28.81 | 36.23 | 32.52 | 41.58 |
| WUDI-Merging | 55.98 | 58.55 | 57.26 | 63.49 | 16.11 | 39.80 | 25.08 | 37.83 | 31.45 | 42.84 |
| w/ HARC | 56.38 | 58.91 | 57.64 | 64.56 | 17.50 | 41.03 | 25.39 | 37.89 | 31.64 | 43.44 |
| Fisher Merging | 44.40 | 59.70 | 52.05 | 68.18 | 16.19 | 42.18 | 16.12 | 33.95 | 25.03 | 39.76 |
| w/ HARC | 44.22 | 59.28 | 51.75 | 68.76 | 16.59 | 42.68 | 16.92 | 34.01 | 25.47 | 39.96 |
| RegMean | 52.36 | 62.24 | 57.30 | 67.69 | 16.88 | 42.28 | 21.34 | 34.90 | 28.12 | 42.57 |
| w/ HARC | 51.57 | 62.37 | 56.97 | 67.57 | 16.95 | 42.26 | 21.85 | 35.68 | 28.76 | 42.67 |

As shown in Table 5, HARC consistently improves the overall score across all six merging baselines. Notably, WUDI+HARC achieves the best overall score of 43.44, a +0.60 improvement over the uncalibrated baseline. These results demonstrate that HARC remains effective even as the routing complexity increases with more models.

## D.2. Cross-Architecture Generalization

To evaluate the generalizability of HARC beyond OLMoE, we conduct experiments on Qwen3-30B-A3B (Yang et al., 2025), a significantly larger MoE model with 128 experts, 8 of which are active per token. We fine-tune separate models on math and code tasks, and evaluate the merged models on the same benchmarks as in Section 5.

As shown in Table 6, HARC consistently improves all five merging strategies, with overall gains ranging from +0.27 (Fisher) to +0.64 (DARE). Notably, WUDI+HARC achieves the best overall score of 75.11, a +0.52 improvement over the uncalibrated baseline. These results confirm that routing breakdown is a structural consequence of the softmax-Top-$k$ mechanism rather than a model-specific artifact: regardless of expert count or model scale, linearly merging router weights perturbs the delicate logit balance. The consistent gains across architectures also validate that our quadratic approximation (Lemma 4.1) and Hessian-based solution (Theorem 4.2) capture a genuine and transferrable property of MoE routing.

*Table 6.* Multi-task performance when merging Qwen3-30B-A3B models on mathematics reasoning and code generation tasks.

| Method | Math | | | Code | | | Overall |
|---|---|---|---|---|---|---|---|
| | GSM8K | MATH500 | Average | HumanEval+ | MBPP+ | Average | |
| Individual | 87.97 | 57.65 | 72.81 | 81.29 | 76.42 | 78.86 | 75.83 |
| Weight Averaging | 87.86 | 56.78 | 72.32 | 80.30 | 76.03 | 78.17 | 75.24 |
| w/ HARC | 87.91 | 56.23 | 72.07 | 81.55 | 76.36 | 78.96 | 75.51 |
| TIES-Merging | 87.64 | 56.14 | 71.89 | 81.10 | 75.23 | 78.17 | 75.03 |
| w/ HARC | 87.76 | 56.98 | 72.37 | 80.76 | 76.05 | 78.41 | 75.39 |
| DARE | 87.65 | 56.45 | 72.05 | 79.15 | 74.19 | 76.67 | 74.36 |
| w/ HARC | 87.61 | 57.35 | 72.48 | 79.85 | 75.18 | 77.52 | 75.00 |
| WUDI-Merging | 87.34 | 55.18 | 71.26 | 80.22 | 75.61 | 77.91 | 74.59 |
| w/ HARC | 87.53 | 56.19 | 71.86 | 80.61 | 76.12 | 78.37 | 75.11 |
| Fisher Merging | 87.95 | 55.38 | 71.67 | 78.35 | 75.94 | 77.15 | 74.41 |
| w/ HARC | 87.99 | 56.94 | 72.47 | 78.58 | 75.38 | 76.98 | 74.72 |

## D.3. Complementarity to Post-Merge Fine-Tuning

A natural question is whether post-merge supervised fine-tuning (SFT) can naturally resolve routing breakdown, potentially obviating the need for HARC. To investigate this, we conduct experiments applying SFT after merging with and without HARC. Specifically, we use the same calibration data for fine-tuning the merged model on both math and code tasks.

*Table 7.* Compatibility of HARC with post-merge SFT. KL Div. denotes the average routing KL divergence between the merged and source routers. **Bold** values in each group indicate the best task performance or the lowest KL Div.

| Method | Math | Code | Overall | KL Div. |
|---|---|---|---|---|
| **WUDI** | 43.44 | 32.73 | 38.08 | 0.00508 |
| +HARC | 44.01 | 33.90 | 38.96 | **0.00484** |
| +SFT | 43.42 | 33.28 | 38.35 | 0.00544 |
| +HARC+SFT | **44.62** | **34.10** | **39.36** | 0.00529 |
| **RegMean** | 43.64 | 30.64 | 37.14 | 0.00456 |
| +HARC | 44.09 | 31.08 | 37.58 | **0.00354** |
| +SFT | 44.17 | 31.18 | 37.67 | 0.00505 |
| +HARC+SFT | **44.24** | **31.39** | **37.81** | 0.00470 |

The results in Table 7 reveal three key findings. **First**, while SFT improves task accuracy, it *increases* the routing KL divergence (e.g., from 0.00508 to 0.00544 for WUDI), demonstrating that standard fine-tuning does not correct the router; instead, it forces the entire model to re-converge to a different local optimum. **Second**, HARC directly *reduces* routing KL divergence, providing targeted structural repair. In scenarios with severe routing breakdown (WUDI), this repair yields substantially larger gains (+0.88) compared to SFT's blind re-convergence (+0.27). **Third**, HARC+SFT achieves the best overall performance (39.36 for WUDI, 37.81 for RegMean), confirming that HARC provides a structurally sound initialization that allows subsequent fine-tuning to focus on task adaptation rather than compensating for routing damage.

