# OpenReview forum: "When Model Merging Breaks Routing: Training-Free Calibration for MoE"
_ICML.cc/2026/Conference — ICML 2026 regular_

### Official Review · Reviewer_Tgmm · 2026-02-19

**Soundness:** 2
**Presentation:** 4
**Significance:** 2
**Originality:** 4
**Overall Recommendation:** 4
**Confidence:** 3

**Summary:**

This paper identifies the problem that existing merging techniques, largely based on linear parameter arithmetic or optimization, struggle when applied to Mixture-of-Experts (MoE) architectures. Then it points out one possible reason, "routing breakdown," which results from the router's nonlinearity.  To address the problem, the authors proposed Hessian-Aware Router Calibration (HARC), a training-free framework that leverages second-order curvature information to merge routers and test the method on math and code generation tasks. Experiments on mathematical reasoning and code generation show that adding HARC consistently improves performance across multiple merging baselines compared with using those baselines alone. The paper also includes ablations and analyses on calibration data composition, data efficiency, and HARC’s effectiveness in reducing cumulative routing deviation across layers.

**Compliance With Llm Reviewing Policy:**

Affirmed.

**Final Justification:**

The authors provide additional analyses, including oracle routing, runtime characterization, and disagreement metrics, which meaningfully clarify several key concerns and strengthen the empirical support for the method. The clarification regarding KL divergence further improves the coherence and persuasiveness of the theoretical justification.

**Key Questions For Authors:**

## The most important questions are

1. What is the practical runtime of HARC, and the comparison between it and baselines?

2. How close is HARC to an oracle router upper bound? Can you isolate how much of the merge degradation comes from the router vs the experts?

3. Can you report token-level routing disagreement rates?

## Personal curiosity, no need to answer

1. What happens if you only calibrate the last N layers where deviation accumulates? This matters for speed.

**Limitations:**

yes

**Strengths And Weaknesses:**

# Strengths

- Soundness: Give a clear motivation by the diagnosis of routing breakdown in MoE merging. HARC is a technically plausible, training-free router calibration approach, and is supported by clear approximation theories and ablation analysis. Empirical gains are also consistent across multiple merge baselines,
- Presentation: Clear end-to-end story with helpful figures that make the paper easy to follow.
- Significance: The paper tackles a practical problem of combining MoE models. The training-free, matrix-free nature of HARC makes it potentially useful for both researchers and practitioners.
- Originality: Novel framing of MoE merging failure as a router-specific instability, and addressing the problem by a second-order, training-free algorithm. The analysis on HARC mitigating routing collapse in sensitive domains and cumulative routing deviation is interesting.

# Weaknesses

## Soundness

1. How close is HARC to an “oracle router” upper bound?

The paper’s central claim is that MoE merging fails primarily due to router issues. Without an oracle-router comparison (or a decomposition that isolates router vs expert effects), it is hard to quantify how much of the overall degradation is truly attributable to router merging, and how much remains due to other factors. I understand you did that in Fig. 1 (which I will also mention in 2), but a further discussion in the experimental section would help.

2. Limited router shift on math tasks and implications for the diagnosis.

If router mismatch is small in some regimes, it weakens the causal narrative that “router breakdown” is the dominant source of performance loss. This raises the possibility that other mechanisms (expert incompatibility, representation drift, etc.) may be comparably important.

3. Computational and memory cost

HARC appears to run sequentially layer-by-layer. Even if it is training-free, the sequential calibration plus Hessian-related computations may be nontrivial. The paper would be stronger with either (i) explicit timing/memory comparisons against baselines, or (ii) a clear complexity discussion tied to actual measured overhead.

4. Is KL divergence the right evidence for routing breakdown

KL can be hard to interpret in MoE routing because small distribution shifts may or may not translate into Top-k changes, and vice versa. It may not be clear how to interpret the change in accuracy from the statistics. A token-level routing consistency/disagreement rate would be more directly aligned with the stated failure mode.

5. Why is the KL improvement small at early layers?

Without direct evidence, the interpretation of “small early-layer KL improvement” is somewhat speculative. Maybe give some analysis or potential reasons.

## Presentation

1. Minor issue: typo in Fig. 3(c)

There appears to be a typo above the "Further Analysis" section referencing Fig. 3(c). Please fix for clarity and consistency.

## Significance

1. Practical overhead vs modest gains

The improvements are consistent but sometimes modest, so practical significance depends on whether the overhead is small relative to the benefit. A clearer cost–benefit characterization would make the method more actionable.

## Originality

1. Strengthening the “router is the bottleneck” story

HARC is conceptually novel as a second-order, training-free router calibration method, but the paper’s originality and insight would be clearer if the experiments more directly validate the underlying hypothesis that router failure is the principal bottleneck in MoE merging (and quantify how much HARC closes that gap).

---

> ### Author Rebuttal · Authors · 2026-03-31
>
> ## Q1: Practical Runtime and Efficiency
>
> HARC is a **one-time offline calibration** that does not affect inference speed. We report runtime and memory below:
>
> | Method | Time (min) | Memory (GB) |
> |:--|:--:|:--:|
> | Data-free (TIES, DARE, WUDI, etc.) | <5 | ~80 |
> | RegMean | 12.4 | 120.9 |
> | HARC | 36.2 | 118.7 |
>
> Memory is comparable to RegMean due to our matrix-free CG solver. The additional time stems from iterative CG convergence. Importantly, HARC supports **partial-layer calibration** for a flexible cost-performance tradeoff:
>
> | Last N Layers | Performance | Est. Time |
> |:--:|:--:|:--:|
> | 0 (Baseline) | 38.08 | - |
> | 4 | 38.40 (+0.32) | ~9 min |
> | 8 | 38.54 (+0.46) | ~18 min |
> | 16 (All) | 38.96 (+0.88) | ~36 min |
>
> Calibrating only the last 4 layers captures **36% of the full gain in approximately 9 minutes**, which is comparable to RegMean's total cost. This makes HARC practical even under tight computational budgets.
>
> ## Q2: Oracle Router Analysis
>
> We conducted an Oracle Router experiment that forces the merged model to use source routing decisions, isolating router vs. expert contributions. Results are broken down by domain to address the concern about math tasks:
>
> | Method | Math | Code | Overall |
> |:--|:--:|:--:|:--:|
> | Individual | 43.37 | 36.73 | 40.05 |
> | WUDI-Merging | 43.44 | 32.73 | 38.08 |
> | w/ HARC | 44.01 | 33.90 | 38.96 |
> | w/ Oracle Router | 44.12 | 34.65 | 39.39 |
>
> **Decomposition:** WUDI suffers a 1.97-point overall drop from `Individual`. The Oracle Router recovers 1.31 points (**66.5%**), confirming router mismatch as the **dominant** degradation source. HARC recovers 0.88 points, closing **67.2%** of the gap to the Oracle Router.
>
> **Domain-specific analysis:** The Oracle Router's gain is larger on Code (+1.92) than Math (+0.68), which is consistent with code generation requiring more precise expert specialization. For math, experts may be more interchangeable, so routing perturbations cause less damage. This domain-dependent sensitivity actually validates HARC's design: its Hessian-based importance weighting (Lemma 1) naturally allocates more calibration effort to routing-sensitive domains. Notably, HARC still improves Math (43.44→44.01).
>
> In summary, these results confirm router mismatch as the principal bottleneck and demonstrate that HARC effectively closes this gap. We will include these experimental results and discussions in the revised manuscript.
>
>
> ## Q3: Token-Level Routing Disagreement Rate
>
> We report the fraction of tokens whose Top-$k$ assignments differ from source routers, averaged across all layers:
>
> | Method | w/o HARC | w/ HARC |
> |:--|:--:|:--:|
> | Weight Averaging | 49.9 | 47.3 |
> | TIES-Merging | 51.2 | 47.6 |
> | DARE | 55.4 | 48.1 |
> | WUDI-Merging | 50.1 | 46.4 |
> | Fisher Merging | 48.0 | 45.3 |
> | RegMean | 46.9 | 43.9 |
>
> HARC consistently reduces disagreement by **2.6–7.3 pp** across all methods.
>
> Regarding the still-high absolute values: in OLMoE ($K$=64, Top-8), many experts encode overlapping functionalities, so a disagreement does not necessarily indicate catastrophic misrouting. Critical failures occur only when tokens reach *functionally dissimilar* experts. HARC's Hessian weighting prioritizes correcting these high-impact cases by assigning higher importance to highly-activated experts (Lemma 1), which explains why moderate disagreement reduction yields meaningful performance gains.
>
> ## Q4: KL Divergence as Evidence
>
> We agree that KL alone does not perfectly capture Top-$k$ changes, which motivated the disagreement rate analysis in Q3. We view the two metrics as **complementary**: KL captures *soft* routing shifts (changes in expert weighting even when the Top-$k$ set is unchanged), while the disagreement rate measures discrete selection outcomes. Together they provide a more complete diagnostic. We will clarify this complementary relationship in the revision.
>
> ## Q5: Small KL Improvement in Early Layers
>
> We point to the quantitative evidence in Figure 4(a): early layers (1–10) exhibit baseline KL divergence below **0.05**, leaving minimal room for improvement. In contrast, deep layers (11–16) show KL values exceeding **0.3**, where HARC provides its most substantial correction.
>
> This pattern reflects that shallower layers learn more general, task-agnostic representations whose routing is less disrupted by merging, while deeper layers encode specialized knowledge with higher routing sensitivity. The partial calibration results in Q1 corroborate this: calibrating only the last 4 layers already captures 36% of the total gain, confirming that deep layers are the critical bottleneck. We will clarify this point in the revision.
>
> ## Q6: Typo
>
> Thank you for pointing this out. The correct reference is Figure 3(b). We will fix this in the revision.

---

> > ### Author Rebuttal · Reviewer_Tgmm · 2026-04-03
> >
> > Thank you for the clear and thorough rebuttal. The additional experiments on runtime analysis, oracle routing, and disagreement rates effectively address my main concerns, and the clarification regarding KL divergence is also helpful. Overall, I am inclined to maintain my weak accept recommendation.

---

### Official Review · Reviewer_zL74 · 2026-03-13

**Soundness:** 3
**Presentation:** 2
**Significance:** 3
**Originality:** 2
**Overall Recommendation:** 4
**Confidence:** 4

**Summary:**

The paper identifies a specific problem in model merging called "routing breakdown," which happens when merging Mixture-of-Experts (MoE) models. Standard linear merging methods fail because MoE routing uses non-linear softmax and top-k operations. To fix this, the authors propose HARC (Hessian-Aware Router Calibration). This is a training-free method that aligns the merged router's distribution with the source routers. It uses a second-order Hessian approximation of the KL divergence and solves it efficiently with a matrix-free conjugate gradient method. The experiments show that HARC improves performance on math and code tasks compared to standard merging baselines.

**Compliance With Llm Reviewing Policy:**

Affirmed.

**Final Justification:**

The authors' rebuttal answers my questions. I am raising my score to weak accept.

1. They tested a bigger model (Qwen3-30B) and merged 3 models. The method generalizes well to more complex settings.
2. They tested the method with noisy and unbalanced data. The performance is robust.
3. They proved that we can just use $\alpha=0.9$ as a default setting. The math solver is stable.

**Key Questions For Authors:**

Q1. You only evaluated HARC on the OLMOE-1B-7B model. Does routing breakdown get worse as the total number of experts increases? How does your method perform on larger or different MoE architectures, such as Mixtral 8x7B?

Q2. Merging math and code is a good starting point. However, what happens when you merge more than two models, or models from very different domains (e.g., medical and legal text)? Does the conjugate gradient solver still converge quickly when the routing distributions of the source models conflict heavily with each other?

Q3. In Figure 3(b), the performance drops sharply for certain values of the regularization factors. How sensitive is the method to $\alpha$ in practice? Do users need a validation set to tune this parameter for every new merge? If extensive tuning is required, it reduces the appeal of a "training-free" method.

Q4. You mention that the matrix-free approach is efficient. Can you provide a concrete comparison of the wall-clock time and peak memory usage? It would be helpful to see a table comparing HARC's resource costs directly against data-driven baselines like RegMean.

**Limitations:**

L1. The reliance on calibration data is a practical limitation. Although Figure 4 shows the method is data-efficient, obtaining high-quality, task-specific prompt data for every source model is not always possible in real-world scenarios. The authors should discuss what happens to the router calibration when the available data is noisy, out-of-domain, or highly imbalanced across the different tasks.

L2. The paper claims to solve MoE merging broadly, but the empirical evidence is limited to a two-task merge. The theoretical formulation clearly allows for $D$ tasks, but without experiments merging 3 or more models, the scalability of the method is a limitation that needs to be openly acknowledged. Interference between routing distributions will likely grow complex as more tasks are added.

L3. The authors briefly mention numerical instability when scaling the diagonal parameter $\gamma$. They should expand on the limits of their conjugate gradient solver. For instance, if the source routing matrices are highly sparse or specialized, could the solver fail to converge or require too many iterations?

**Strengths And Weaknesses:**

**Strengths:**
- Pointing out that routing breakdown causes MoE merging to fail is a valuable and novel observation.
- The math behind the quadratic approximation is solid. Using a matrix-free solver to handle the large dimensions of the Hessian is a smart engineering choice.
- The method is highly practical. It acts as a plug-and-play step that improves multiple existing merging baselines (like TIES and WUDI).

**Weaknesses:**
- The paper only tests one model family (OLMOE-1B-7B) on two domains (math and code). It is not clear if this method works well on larger MoE models.
- Real-world merging often involves more than two models. The paper lacks experiments merging three or more experts.
- The paper shows that the method is sensitive to the regularization parameters, but it does not give a clear guide on how to pick these parameters for new, unseen tasks.

---

> ### Author Rebuttal · Authors · 2026-03-31
>
> ## Q1: Generalizability across Architectures
>
> We thank the reviewer for raising this important point. To address it, we conducted experiments on Qwen3-30B-A3B, a significantly larger architecture than OLMoE in our main experiments. The results show that:
>
> * Due to the increased number of experts (128 vs. 64), the routing disagreement rate after applying WUDI-Merging is indeed slightly higher compared to OLMoE (52.52% vs. 50.11%).
> * HARC effectively mitigates this routing breakdown on Qwen as well, improving performance across various merging methods. To dedicate sufficient space to address your other insightful comments in depth, we refer you to our response to QFHY Q2 for the detailed quantitative results.
>
> These results confirm that HARC generalizes robustly to larger MoE architectures.
>
> ## Q2: Merging More Models
>
> To evaluate the scalability of HARC, we extended our setup to merge a third model (a chat model). Results show:
>
> * **Performance**: HARC consistently improves the overall performance across various baseline methods, demonstrating its effectiveness in multi-model merging scenarios. Detailed results can be found in our response to Reviewer Tg1c (Q1), who also highlighted the importance of this scalability test.
>
> * **CG convergence**: Despite a slight increase in routing disagreement (50.1% → 51.4%), the average CG iterations actually decreased (256.1 → 231.9). This acceleration is driven by the complementary null spaces of the domain-specific precision matrices. Specifically, the positive eigenvalues of $\Sigma_3$ fill the near-zero eigenvalues of $\Sigma_1$ and $\Sigma_2$. This raises the minimum eigenvalue of the aggregate matrix $\Sigma=\sum_i\Sigma_i$, thereby reducing its condition number and accelerating CG convergence.
>
> These findings confirm that HARC scales robustly to multi-model merging scenarios. We will add these results to the revised manuscript.
>
> ## Q3: Sensitivity of the Regularization Parameter
>
> To directly test sensitivity on an unseen scenario, we conducted a new 3-model WUDI merging experiment:
>
> | $\alpha$ | Performance |
> | :-: | :-: |
> | 1.0 | 42.76 |
> | 0.95 | 42.87 |
> | 0.9 | 43.44 |
> | 0.85 | 43.00 |
> | 0.8 | 42.85 |
>
> While regularization impacts absolute performance, $\alpha=0.9$ hits the peak. This generalizability is consistent with our main results (Table 1), where fixing $\alpha=0.9$ across **all six** diverse merging baselines yielded consistent improvements without any tuning. Thus, $\alpha=0.9$ serves as a **robust default** for most scenarios, which eliminates the need for hyperparameter tuning. We will clarify this in the revised manuscript.
>
> ## Q4: Efficiency
>
> To clarify, our "efficient" claim refers to avoiding the prohibitive cost of explicitly constructing the precision matrix. However, we acknowledge that the iterative nature of the matrix-free CG solver results in higher wall-clock time compared to closed-form methods like RegMean.
>
> As shown in the table below, HARC maintains a comparable memory footprint to RegMean by avoiding materializing the large Hessian matrix. The increased time cost stems from the multiple iterations required by the CG algorithm to apply the linear operator over cached hidden states. In future work, we plan to incorporate a lightweight diagonal preconditioner to accelerate the CG solver and narrow this time gap.
>
> | Method | Time (min) | Memory (GB) |
> |:-|:-:|:-:|
> | RegMean | 12.4 | 120.9 |
> | HARC | 36.2 | 118.7 |
>
> We will include these discussions in the revised manuscript.
>
> ## Q5: Reliance on Calibration Data
>
> We would like to clarify that relying on calibration data is a shared characteristic of all data-driven merging methods, not a risk unique to HARC. To systematically investigate this issue under non-ideal conditions, we conducted additional experiments:
>
> * **Noisy and OOD Data:** We randomly sampled 10,000 general texts from C4-en and assigned them to source models based on language modeling loss. Despite using this noisy, OOD data, HARC retained **42%** of the optimal performance gain.
> * **Highly Imbalanced Data:** Building on the natural data imbalance (math:code ≈ 2:1) of our main experiment, we reduced code data to 10% (creating a harsh 20:1 imbalance). HARC still maintained **58%** of the original improvement.
>
> | Method | Score |
> | :- | :- |
> | WUDI-Merging | 38.08 |
> | w/ HARC (SrcCorrect) | 38.96 ($\uparrow$ 0.88) |
> | w/ HARC (C4) | 38.45 ($\uparrow$ 0.37) |
> | w/ HARC (Imbalance) | 38.59 ($\uparrow$ 0.51) |
>
> These results demonstrate HARC's robustness to data quality and distribution skew. We will add this analysis to the revised manuscript.
>
> ## Q6: Convergence Limits of the CG Solver
>
> While sparse routing can render $\Sigma$ ill-conditioned, our diagonal regularization explicitly mitigates this by lifting near-zero eigenvalues to reduce the condition number. Empirically, the CG solver consistently converged (residual $<10^{-6}$ within 199–364 iterations) across all experiments. We will clarify this in the revision.

---

> > ### Author Rebuttal · Reviewer_zL74 · 2026-04-03
> >
> > My concerns have been adequately addressed.

---

> > > ### Author Response · Authors · 2026-04-03
> > >
> > > Thank you for acknowledging our rebuttal—we’re glad it addressed your concerns. We would appreciate it if you could consider revisiting the score accordingly.

---

### Official Review · Reviewer_Tg1c · 2026-03-13

**Soundness:** 3
**Presentation:** 3
**Significance:** 3
**Originality:** 3
**Overall Recommendation:** 4
**Confidence:** 4

**Summary:**

see below

**Compliance With Llm Reviewing Policy:**

Affirmed.

**Final Justification:**

I have carefully read the rebuttal by authors. I will keep my score of 4.

**Key Questions For Authors:**

**Main Contributions:**

The paper identifies and systematically analyzes a critical failure mode in merging Mixture-of-Experts (MoE) models: "routing breakdown." It demonstrates that standard model merging methods, which assume linear mode connectivity, fail for MoE architectures because the non-linear routing mechanism (softmax + Top-k) is highly sensitive to parameter perturbations.



The paper then provides empirical evidence that this leads to catastrophic expert misassignment (over 50% deviation) and proposes Hessian-Aware Router Calibration (HARC), a training-free method that aligns the merged router's output distribution with the source models. The idea is to  use a second-order approximation and a matrix-free conjugate gradient solver.



**Strengths:**

The problem is well-motivated and timely. The proposed solution seems reasonable. The writing and presentation is mostly clear. The experimental results are consistent and show intriguing improvements across multiple baselines.

**Weaknesses (and my major concerns):**

1. Correct me if wrong: The paper's scope is limited to merging only two models (math and code). While this is a clean setup for analysis, the scalability to merging more than two models with potentially conflicting routing behaviors is not explored.  The paper would be stronger with at least a small-scale experiment merging three models.

2. The theoretical derivation relies on a second-order Taylor expansion of the log-partition function. This approximation is accurate only when the merged logits z_m are close to the source logits z_i. However, in practice, the merged router can be quite different from the source routers, especially after aggressive merging. The paper does not discuss the regime of validity for this approximation or provide diagnostics (e.g., checking the norm of (z_m - z_i) to ensure the quadratic approximation is reasonable.

3. The computational cost of HARC is not well discussed. How much overhead is required for conjugate gradient steps?





4. On the Top-k selection: The derivation in Lemma 1 uses the full softmax distribution, but inference uses discrete Top-k selection. Is minimizing KL divergence over the full distribution the right proxy for aligning Top-k decisions? More discussion is needed here.



**Presentation suggestions**

5. For Eq. (5), please specify the optimization objective, and the optimization variable (rm or Wm).

**Limitations:**

see above

**Strengths And Weaknesses:**

see below

---

> ### Author Rebuttal · Authors · 2026-03-31
>
> ## Q1: Merging More Models
>
> We appreciate the reviewer's constructive suggestion to explore the scalability of HARC. To address this, we conducted a three-model merging experiment by adding a chat model to the existing setup.
>
> Results show that HARC consistently improves performance across all merging baselines, even as the routing complexity increases with more models:
>
> |Method| Chat | Math | Code | Overall |
> |:-|:-:|:-:|:-:|:-:|
> | Individual | 61.02 | 43.37 | 36.73 | 47.04 |
> | Weight Averaging | 52.77 | 41.34 | 26.41 | 40.17 |
> | &nbsp;&nbsp; w/ HARC | **52.85** | **42.06** | **26.90** | **40.60** |
> | TIES-Merging | 55.22 | 41.23 | 26.73 | 41.06 |
> | &nbsp;&nbsp;w/ HARC  | **55.28** | **40.92** | **27.80** | **41.33** |
> | DARE | 50.54 | 40.49 | **32.68** | 41.24 |
> | &nbsp;&nbsp;w/ HARC  | **51.13** | **41.09** | 32.52 | **41.58** |
> | WUDI-Merging | 57.26 | 39.80 | 31.45 | 42.84 |
> | &nbsp;&nbsp;w/ HARC  | **57.64** | **41.03** | **31.64** | **43.44** |
> | Fisher Merging | **52.05** | 42.18 | 25.03 | 39.76 |
> | &nbsp;&nbsp;w/ HARC  | 51.75 | **42.68** | **25.47** | **39.96** |
> | RegMean | **57.30** | **42.28** | 28.12 | 42.57 |
> | &nbsp;&nbsp;w/ HARC  | 56.97 | 42.26 | **28.76** | **42.67** |
>
> We will include these experimental results and discussions in the revised manuscript.
>
> ## Q2: Validity of the Quadratic Approximation
>
> To clarify, the approximation holds well in practice because the divergence between merged and source logits is minimized by the merging process itself:
>
> * For arithmetic-based methods (e.g., TIES, DARE), the logit deviation $\mathbf{z}_m - \mathbf{z}_i = (\mathbf{W}_m - \mathbf{W}_i)\mathbf{x}$ is determined by the norm of task vectors, which are typically small in fine-tuning settings.
> * Optimization-based methods either explicitly minimize $||\mathbf{z}_m - \mathbf{z}_i||$ (e.g., RegMean) or implicitly constrain this deviation during interference reduction (e.g., WUDI), ensuring $z_m$ remains close to $z_i$
>
> To empirically validate this, we calculated the relative logits shift $\frac{||\mathbf{z}_m - \mathbf{z}_i||}{||\mathbf{z}_i||}$ on calibration data, finding that for TIES-Merging, 93.1% of tokens (90.8% for WUDI-Merging) exhibit a relative shift of less than 0.1, and 99.4% (99.3% for WUDI) exhibit a shift of less than 0.2. These statistics confirm that the small-perturbation assumption holds, justifying the theoretical derivation.
>
> ## Q3: Computational Overhead
>
> We acknowledge that the overhead of the conjugate gradient steps should be better discussed. Notably, the efficient computation method in Section 4.3 reduces both spatial and time complexity.
>
> * **Before optimization:** Explicitly constructing the aggregate precision matrix $\mathrm{\Sigma}$ takes $O(NK^2d^2)$, and directly inverting it requires $O(K^3d^3)$. The total time complexity is $O(NK^2d^2+K^3d^3)$, which is computationally prohibitive when the dimension $d$ is large (e.g., $d=2048$ in OLMoE-1B-7B).
> * **After optimization:** By avoiding explicit matrix construction and inversion, the key step of our algorithm relies on the operator $A(\mathrm{w})$ defined in Equation 13, which only takes $O(Kd+K^2)$. For $N$ samples and $T$ conjugate gradient iterations, the overall time complexity is reduced to $O(TN(Kd+K^2))$.
>
> To provide a concrete measure of the actual overhead required for the conjugate gradient steps, we conducted practical runtime evaluations. In our experiments, we found that the total elapsed time of HARC is approximately 3 times that of RegMean, which we consider a very reasonable trade-off for the performance gains.
>
> | Method  | Elapsed Time (min) |
> | :-| :-: |
> | RegMean | 12.4 |
> | HARC | 36.2 |
>
> We will add this complexity analysis to the revised manuscript to clarify the computational overhead.
>
> ## Q4: Optimization over Full Softmax Distribution
>
> To clarify, directly optimizing discrete Top-k selection is intractable because its step-function nature yields zero gradients when expert rankings change, necessitating a differentiable proxy. As analyzed in the interpretation of Lemma 1, minimizing the full-distribution KL divergence serves as an ideal proxy because matching softmax probabilities inherently preserves the strict relative logit rankings required for Top-k. Specifically, the Hessian weighting (where diagonal terms scale with routing probability) heavily penalizes perturbations for active (Top-k) experts while tolerating noise from unselected ones. Crucially, the off-diagonal terms capture the competitive zero-sum dynamics of softmax, ensuring that aligning the continuous distribution reliably prevents discrete ranking flips.
>
> ## Q5: Mathematical Notation
>
> Thank you for raising this point. We have revised Equation (5) to explicitly formulate it as an optimization problem:
>
> $$
> \mathbf{W} _ m^* = {\arg\min} _ {\mathbf{W} _ m} \sum _ {i=1}^D \mathbb{E} _ {\mathbf{x} \sim \mathcal{D} _ i} [\operatorname{KL}(\mathbf{r} _ i(\mathbf{x}) \parallel \mathbf{r} _ m(\mathbf{x}; \mathbf{W} _ m))]
> $$

---

> > ### Author Rebuttal · Reviewer_Tg1c · 2026-04-03
> >
> > Thanks for the rebuttal! I will keep my postive score and vote for acceptance.

---

### Official Review · Reviewer_QFHY · 2026-03-15

**Soundness:** 2
**Presentation:** 2
**Significance:** 2
**Originality:** 3
**Overall Recommendation:** 3
**Confidence:** 4

**Summary:**

This paper addresses a critical failure mode in merging Mixture-of-Experts (MoE) large language models, termed "routing breakdown." The authors identify that standard linear parameter merging techniques (e.g., Task Arithmetic, TIES-Merging) disrupt the non-linear softmax and discrete Top-k routing mechanisms, leading to severe expert mismatch and performance degradation, particularly in deeper layers. To mitigate this, the paper proposes Hessian-Aware Router Calibration (HARC), a training-free framework that leverages second-order curvature information to realign the merged router's output distribution with that of the source models. HARC formulates the alignment as a quadratic optimization problem solvable via a matrix-free conjugate gradient method. Experiments on mathematical reasoning and code generation tasks using OLMoE models demonstrate that HARC significantly improves performance across various merging baselines by preserving correct expert selection logic.

**Compliance With Llm Reviewing Policy:**

Affirmed.

**Final Justification:**

The authors’ rebuttal and additional SFT experiments partially address my concerns by confirming HARC’s utility in lightweight fine-tuning regimes. However, the evaluation remains limited to low-data scenarios, leaving it unclear whether HARC offers significant advantages over standard merging when models undergo standard fine-tuning where optimizers might naturally correct routing errors. I maintain my recommendation of Weak Rejection.

**Key Questions For Authors:**

- Post-Merge Fine-Tuning: Have you investigated whether the performance degradation caused by routing breakdown can be recovered through standard post-merge fine-tuning? If a merged model (without HARC) is fine-tuned on a small mixed dataset, does it converge to a similar performance level as a HARC-calibrated model, or does HARC provide a superior initialization that leads to better final convergence or faster training?

- Generalizability across Architectures: The experiments are currently limited to the OLMoE architecture. Can you discuss or provide evidence on how HARC performs on MoE models with significantly different structures?

**Limitations:**

The authors have partially discussed limitations regarding data composition and the assumption of local linearity for the Taylor expansion. However, they have not adequately discussed the limitation regarding the interaction with post-merge fine-tuning workflows or the lack of validation across diverse MoE architectures beyond OLMoE. I suggest adding a discussion on whether HARC serves as a replacement for fine-tuning or a complementary initialization strategy, and explicitly acknowledging the need for future work to validate the method on a broader range of MoE models to ensure generalizability.

**Strengths And Weaknesses:**

Strengths:

- Significance and Originality: The identification of "routing breakdown" as a distinct failure mode in MoE merging is a valuable insight. Overall, a major problem investigated by this article is the incompatibility of linear weight averaging with the non-linear, discrete nature of MoE gating mechanisms. The proposal to address this via second-order calibration rather than simple parameter interpolation is novel and well-motivated.
- Soundness: The theoretical derivation, approximating the KL divergence of routing distributions using a Hessian-based quadratic form, is rigorous. The connection to Fisher Information and the justification for capturing off-diagonal terms (competitive dynamics between experts) are convincing.


Weaknesses:

- Interaction with Post-Merge Fine-Tuning: A significant limitation is the scope of the evaluation, which focuses exclusively on training-free merging. In many practical workflows, model merging is followed by a brief period of fine-tuning (e.g., on a mixed dataset) to recover performance or adapt to a new domain. It remains unclear whether the severe performance drops caused by routing breakdown can be naturally alleviated by such subsequent fine-tuning. If standard fine-tuning can easily correct the router weights, the necessity of a complex, training-free calibration step like HARC diminishes. The paper would benefit from a discussion or a small-scale experiment analyzing the convergence behavior of merged MoE models with and without HARC initialization during a short fine-tuning phase.

- Limited Model Diversity: The experimental validation is confined to a single model family (OLMoE). Given the diversity in MoE architectures (e.g., different numbers of experts, varying top-k values), the generalizability of the findings is uncertain. The variance in results, particularly on code generation benchmarks, suggests sensitivity to specific model configurations. Validating the method on other prominent MoE architectures (e.g., Mixtral, Qwen-MoE) is necessary to confirm that "routing breakdown" is a universal phenomenon and that HARC is robust across different structural designs.

- Sensitivity to Calibration Data: While the paper claims data efficiency, the method still relies on a calibration dataset to estimate the Hessian and input covariance. The performance gap between using "SrcCorrect" (filtered correct generations) and noisy data implies that the quality of calibration data is crucial. In scenarios where high-quality, task-specific data is scarce, the robustness of HARC needs further clarification.

---

> ### Author Rebuttal · Authors · 2026-03-31
>
> ## Q1: Necessity of HARC over Post-Merge SFT
>
> We thank the reviewer for this suggestion. We conducted experiments explicitly testing whether standard SFT can replace HARC:
>
> | Method | Math | Code | **Overall** | KL Div. |
> | :- | :-: | :-: | :-: | :-: |
> | **WUDI** | 43.44 | 32.73 | 38.08 | 0.00508 |
> | +HARC | 44.01 | 33.90 | 38.96 | **0.00484** |
> | +SFT | 43.42 | 33.28 | 38.35 | 0.00544 |
> | +HARC+SFT | **44.62** | **34.10** |  **39.36** | 0.00529 |
> | **RegMean** | 43.64 | 30.64 | 37.14 | 0.00456 |
> | +HARC | 44.09 | 31.08 | 37.58 | **0.00354** |
> | +SFT | 44.17 | 31.18 | 37.67 | 0.00505 |
> | +HARC+SFT | **44.24** | **31.39** | **37.81** | 0.00470 |
>
> The results directly validate HARC's necessity:
> 1. **SFT fails to fix routing breakdown.** While SFT improves task accuracy, it *increases* the routing KL divergence. This proves SFT does not correct the router; instead, it forces the entire model to re-converge to a different local optimum.
> 2. **HARC effectively repairs the router.** Unlike SFT, HARC directly reduces the routing KL divergence. In scenarios with severe routing breakdown (WUDI), this targeted structural repair yields much *larger* performance gains (+0.88) compared to SFT's blind re-convergence (+0.27).
> 3. **HARC provides a superior structural initialization.** HARC+SFT achieves the best overall score (39.36), confirming that HARC provides a structurally sound foundation that allows subsequent fine-tuning to focus on task adaptation rather than repairing routing damage.
>
> In short, standard fine-tuning cannot directly fix routing breakdown, whereas HARC not only fixes it but also provides a superior starting point for subsequent fine-tuning. We will include these experimental results and discussions in the revised manuscript.
>
> ## Q2: Generalizability Across Architectures
>
> We would like to clarify that **routing breakdown is a structural consequence of the softmax-Top-$k$ gating mechanism, not an artifact of specific models.** As shown in Section 3.2, any router applying softmax normalization followed by discrete Top-$k$ selection exhibits non-linear sensitivity to parameter perturbations. Since HARC depends only on this structure (the Hessian of the log-partition function and input covariance), it is applicable across MoE configurations without modification.
>
> We extended our evaluation to Qwen3-30B-A3B, which is significantly larger and structurally different from OLMoE-1B-7B in our main experiments:
>
> | Method           | Math       | Code       | **Overall** |
> | :- | :-: | :-: | :-: |
> | Individual       | 72.81      | 78.85      |       75.83       |
> | Weight Averaging | **72.32** | 78.16      |       75.24       |
> | w/ HARC          | 72.07      | **78.96** |  **75.51**  |
> | TIES-Merging     | 71.89      | 78.16      |       75.03       |
> | w/ HARC          | **72.37** | **78.41** |  **75.39**  |
> | DARE             | 72.05      | 76.67      |       74.36       |
> | w/ HARC          | **72.48** | **77.52** |  **75.00**  |
> | WUDI             | 71.26      | 77.92      |       74.59       |
> | w/ HARC          | **71.86** | **78.37** |  **75.11**  |
> | Fisher Merging   | 71.66      | **77.15** |       74.41       |
> | w/ HARC          | **72.46** | 76.98      |  **74.72**  |
>
> HARC improves the overall score on all five merging strategies (+0.27 to +0.64), confirming its generalizability to a new, larger architecture.
>
> ## Q3: Sensitivity to Calibration Data
>
> To address the concern about scenarios where high-quality, task-specific data is scarce, we highlight the following results:
>
> 1. **Unlabeled prompts suffice.** As shown in Figure 3(c) of the paper, HARC is remarkably robust even without filtered, high-quality responses. Using only task-related prompts, HARC achieves 91% of the performance gain observed with the optimal "SrcCorrect" data.
> 2. **General data also works.** To further test HARC under extreme data scarcity, we sample 10,000 text segments from the C4-en general pre-training dataset and associate them with source models based on language modeling loss. Even with this general data, HARC retains 42% of the performance gain observed with the optimal data.
>
> | Method                 | Score                     |
> | :- | :- |
> | WUDI-Merging | 38.08                     |
> | w/ HARC (SrcCorrect)   | 38.96 (+0.88) |
> | w/ HARC (Prompt)       | 38.88 (+0.80) |
> | w/ HARC (C4)           | 38.45 (+0.37) |
>
> These results demonstrate that while high-quality data is beneficial, HARC provides consistent gains across a wide spectrum of data availability and quality. We will include this discussion and the C4 experiment in the revised manuscript.

---

> > ### Author Rebuttal · Reviewer_QFHY · 2026-04-05
> >
> > My mainly concerns have been addressed and I've increased the score.

---

> > > ### Author Response · Authors · 2026-04-07
> > >
> > > We thank the reviewer for acknowledging that the previous concerns were addressed. We respectfully address the remaining question in the *final justification* about whether HARC's advantages persist under standard fine-tuning with more data.
> > >
> > > * **Routing breakdown is a structural problem that more data does not resolve.** The core issue we identify is that softmax+Top-k gating creates non-smooth decision boundaries where small parameter perturbations cause discrete changes in expert selection. Our experiments in Q1 provide direct empirical evidence: SFT *increases* routing KL divergence (WUDI: 0.00508→0.00544; RegMean: 0.00456→0.00505) despite improving task accuracy. This means that SFT, regardless of data scale, adapts expert parameters to work around the broken router rather than correcting the routing itself. More data helps the experts compensate more effectively, but the underlying routing misalignment persists. This is precisely why HARC+SFT consistently outperforms SFT alone: HARC resolves the structural misalignment first, allowing SFT to focus on productive task adaptation.
> > > * **The value of model merging diminishes if extensive fine-tuning is required.** We would also like to note that the central motivation for model merging is to avoid the cost of joint training or extensive fine-tuning. If post-merge fine-tuning with abundant data were always available, one could simply fine-tune a single base model on the combined dataset, bypassing merging entirely. HARC's training-free nature preserves the practical advantage of the merging paradigm while addressing its key failure mode for MoE architectures.
> > > * **HARC's contribution is multi-faceted.** Beyond the training-free setting, our work (1) identifies and characterizes routing breakdown as a previously unrecognized failure mode in MoE merging, (2) provides a principled theoretical framework connecting softmax curvature to merging error (Lemma 1, Theorem 1), and (3) demonstrates consistent improvements across six diverse merging baselines (Table 1), both as a standalone solution and as a complementary initialization for fine-tuning. We believe these contributions are valuable to the community regardless of the downstream fine-tuning regime.
> > >
> > > We hope these clarifications address the reviewer's remaining concern and would be grateful for any reconsideration.

---

### Decision · Program_Chairs · 2026-04-30

**Decision:**

Accept (regular)

**Comment:**

This paper identifies "routing breakdown" in MoE model merging and proposes HARC, a training-free Hessian-aware router calibration method. Initial concerns regarding necessity versus post-merge fine-tuning, limited architecture coverage, and computational overhead were substantially resolved through additional experiments on Qwen3-30B-A3B, three-model merging, and runtime analysis with partial-layer calibration. Following the rebuttal, two reviewers raised their scores, while one maintained Weak Reject citing residual concerns about whether HARC's advantages persist under standard fine-tuning with abundant data. Given the majority positive sentiment, novel identification of the routing breakdown phenomenon, and strong empirical validation across diverse settings, the AC recommend Weak Accept.